# An ErbB2/c-Src axis links bioenergetics with PRC2 translation to drive epigenetic reprogramming and mammary tumorigenesis

Harvey W. Smith et al.[#]

Dysregulation of histone modifications promotes carcinogenesis by altering transcription. Breast cancers frequently overexpress the histone methyltransferase EZH2, the catalytic subunit of Polycomb Repressor Complex 2 (PRC2). However, the role of EZH2 in this setting is unclear due to the context-dependent functions of PRC2 and the heterogeneity of breast cancer. Moreover, the mechanisms underlying PRC2 overexpression in cancer are obscure. Here, using multiple models of breast cancer driven by the oncogene ErbB2, we show that the tyrosine kinase c-Src links energy sufficiency with PRC2 overexpression via control of mRNA translation. By stimulating mitochondrial ATP production, c-Src suppresses energy stress, permitting sustained activation of the mammalian/mechanistic target of rapamycin complex 1 (mTORC1), which increases the translation of mRNAs encoding the PRC2 subunits Ezh2 and Suz12. We show that Ezh2 overexpression and activity are pivotal in ErbB2-mediated mammary tumourigenesis. These results reveal the hitherto unknown c-Src/mTORC1/PRC2 axis, which is essential for ErbB2-driven carcinogenesis.

The lysine methyltransferase Enhancer of Zeste Homolog 2 (EZH2) is the catalytic subunit of Polycomb Repressor Complex 2 (PRC2), which represses transcription via di- and tri-methylation of histone H3 on lysine 27 (H3K27me$^{2/3}$)[1]. A key regulator of pluripotency and differentiation[2,3], PRC2 also has a complex role in cancer, with gain and loss-of-function mutations in PRC2 components reflecting context-specific oncogenic and tumor suppressive activities[4]. Some cancers also overexpress PRC2 components[4], increasing the likelihood that PRC2 will bind chromatin at transiently repressed loci to enforce stable transcriptional silencing[4]. Thus, under conditions where PRC2 exerts oncogenic effects, its overexpression may facilitate tumorigenesis in the absence of activating mutations of its components. This is thought to be the case in breast cancer, where EZH2 overexpression is associated with poor prognosis[5]. However, there are conflicting reports as to whether EZH2 overexpression drives breast cancer growth and progression[5] or is merely correlative[6,7]. This is likely to reflect the heterogeneous nature of breast cancer, which comprises a spectrum of subtypes distinguished by pathological characteristics, gene expression patterns and genomic events[8]. Approximately 15–20% of breast cancers overexpress the receptor tyrosine kinase (RTK) ErbB2/HER2, typically due to *ERBB2* gene amplification[8,9], which leads to aggressive disease with poor prognosis. Despite the fact that ERBB2-positive (ERBB2+) tumors are among those with the highest EZH2 expression and H3K27 tri-methylation[10], few studies have examined the functional requirement for EZH2 specifically in ERBB2+ breast cancer. Crucially, the molecular pathways promoting EZH2/PRC2 overexpression are incompletely understood, despite their potential importance in mediating epigenetic dysregulation and tumor progression in these cancers.

Metabolic reprogramming fuels neoplastic growth by providing energy and biosynthetic intermediates[11] and is also linked to epigenetic dysregulation[12,13] since DNA and histone modifying enzymes require metabolites as cofactors and co-substrates and are thus regulated by pathways producing and consuming these metabolites[12,13]. Metabolic pathways are reprogrammed in cancer cells by genetic alterations in enzymes and their regulators and by aberrant signaling following activation of canonical oncogenes and inactivation of tumor suppressor genes[11]. Although research in this area has focussed largely on up-regulation of aerobic glycolysis (the Warburg Effect), cancer cells can also depend on ATP synthesis through mitochondrial oxidative phosphorylation (OXPHOS) to fulfill their bioenergetic requirements[11]. Given the central role of these metabolic processes in carcinogenesis, further study of their intersections with signaling and epigenetics is warranted.

Here, we apply an integrative approach involving multiple pre-clinical models and analysis of clinical samples to delineate a pathway mediating the overexpression of key PRC2 subunits in ERBB2+ breast cancer. Our results demonstrate a previously unrecognized mechanism whereby the tyrosine kinase c-Src, which is frequently hyper-activated in ERBB2+ breast cancer[14], enhances mitochondrial ATP synthesis to alleviate cellular energy stress. This enables mTORC1 activation, elevating the translation of mRNAs encoding Ezh2 and Suz12, a second essential subunit of PRC2. We show that down-regulation of Ezh2 expression or inhibition of its methyltransferase activity severely impairs the growth of ErbB2+ tumor cells, while inhibition or genetic ablation of Ezh2 in vivo ablates ErbB2-driven mammary epithelial tumorigenesis. Collectively, these observations show how oncogene-dependent bioenergetic modulation, through reprogramming mRNA translation, drives epigenetic alterations that are essential for ErbB2-driven breast cancer.

## Results

### c-Src ablation impairs ErbB2-driven mammary tumorigenesis.
c-Src mediates signaling towards mitogenic and pro-invasive pathways by ErbB2 and related RTKs[15,16]. However, the requirement for c-Src in ErbB2-driven transformation in vivo is unknown. To directly address this issue, we generated a unique GEMM combining conditional *Src* gene targeting (*c-Src^{L/L}*)[17] with mammary epithelial co-expression of oncogenic ErbB2 and Cre recombinase[18] (referred to as NIC; Supplementary Fig. 1a). Ablation of one (*c-Src^{+/L}*) or both (*c-Src^{L/L}*) *Src* alleles significantly delayed mammary tumorigenesis, with the most severe phenotype in the latter (Fig. 1a). *NIC/c-Src^{L/L}* tumors were devoid of c-Src protein (Supplementary Fig. 1b). While c-Src-deficient tumors remained multifocal, their growth was severely impaired (Fig. 1b), correlating with significantly reduced proliferation (Ki67) and impaired cell cycle progression (BrdU) (Fig. 1c). c-Src-deficient tumors retained the solid adenocarcinoma pathology typically associated with ErbB2-expressing GEMMs but showed histological evidence of necrosis (Supplementary Fig. 1c) and slightly increased apoptosis (TUNEL; Supplementary Fig. 1d) as compared to their *c-Src^{+/+}* counterparts. Cells derived from c-Src-deficient tumors proliferated at a dramatically lower rate than c-Src-proficient cell lines in culture (Supplementary Fig. 1e), suggesting that the effects of c-Src deletion on growth are tumor cell-intrinsic. However, as Src family kinase (SFK) activity has been associated with angiogenesis during development and in cancer[19,20], we examined the presence of CD31+ endothelial cells in control and c-Src-deficient tumors, finding no significant difference (Supplementary Fig. 1f). Overall, these data show that c-Src loss causes a tumor cell-intrinsic proliferation defect that significantly impairs ErbB2-driven mammary tumorigenesis.

### c-Src enhances *Ezh2* and *Suz12* translation and PRC2 activity.
To gain mechanistic insight, we determined the effects of c-Src ablation on steady-state mRNA levels in ErbB2-driven mammary tumors using DNA microarrays, identifying 602 transcripts up-regulated and 785 down-regulated in *NIC/c-Src^{L/L}* tumors compared to *NIC/c-Src^{+/+}* controls. Bioinformatic analysis using two independent databases (ChEA and ENCODE) revealed that genes up-regulated in c-Src-deficient tumors were significantly enriched in predicted targets of PRC2, which was the only factor identified using both databases (Fig. 1d). In contrast, the same analyses failed to identify candidate regulators of genes downregulated in c-Src-deficient tumors. KEGG (Kyoto Encyclopedia of Genes and Genomes) and GO (Gene Ontology) analysis revealed that genes up-regulated in c-Src-deficient tumors were involved in adhesion to the extracellular matrix (ECM), cell–cell contact and central carbon metabolism (Supplementary Fig. 2a). While these analyses found fewer pathways and functions among genes down-regulated in *NIC/c-Src^{L/L}* tumors, they did identify down-regulation of protein trafficking pathways and functions typically associated with neurons (Supplementary Fig. 2b).

To confirm our findings, we performed RNA-Seq on cell lines derived from control and c-Src-deficient ErbB2+ tumors and identified candidate regulators of differentially expressed genes using ChEA, ENCODE and ingenuity pathway analysis (IPA), focussing on regulators identified by at least two methods in both tumors and cell lines. Using these criteria, we identified two PRC2 subunits (Ezh2 and Suz12) as candidate regulators of genes up-regulated following c-Src loss (Supplementary Fig. 2c). Accordingly, a significant reduction in H3K27me$^3$ was observed upon c-Src ablation (Fig. 1e), which was confirmed by chromatin immunoprecipitation followed by high-throughput sequencing (ChIP-Seq) (Fig. 1f). KEGG, REACTOME, and GO analyses of genes associated with H3K27me$^3$ converged on neuron-specific

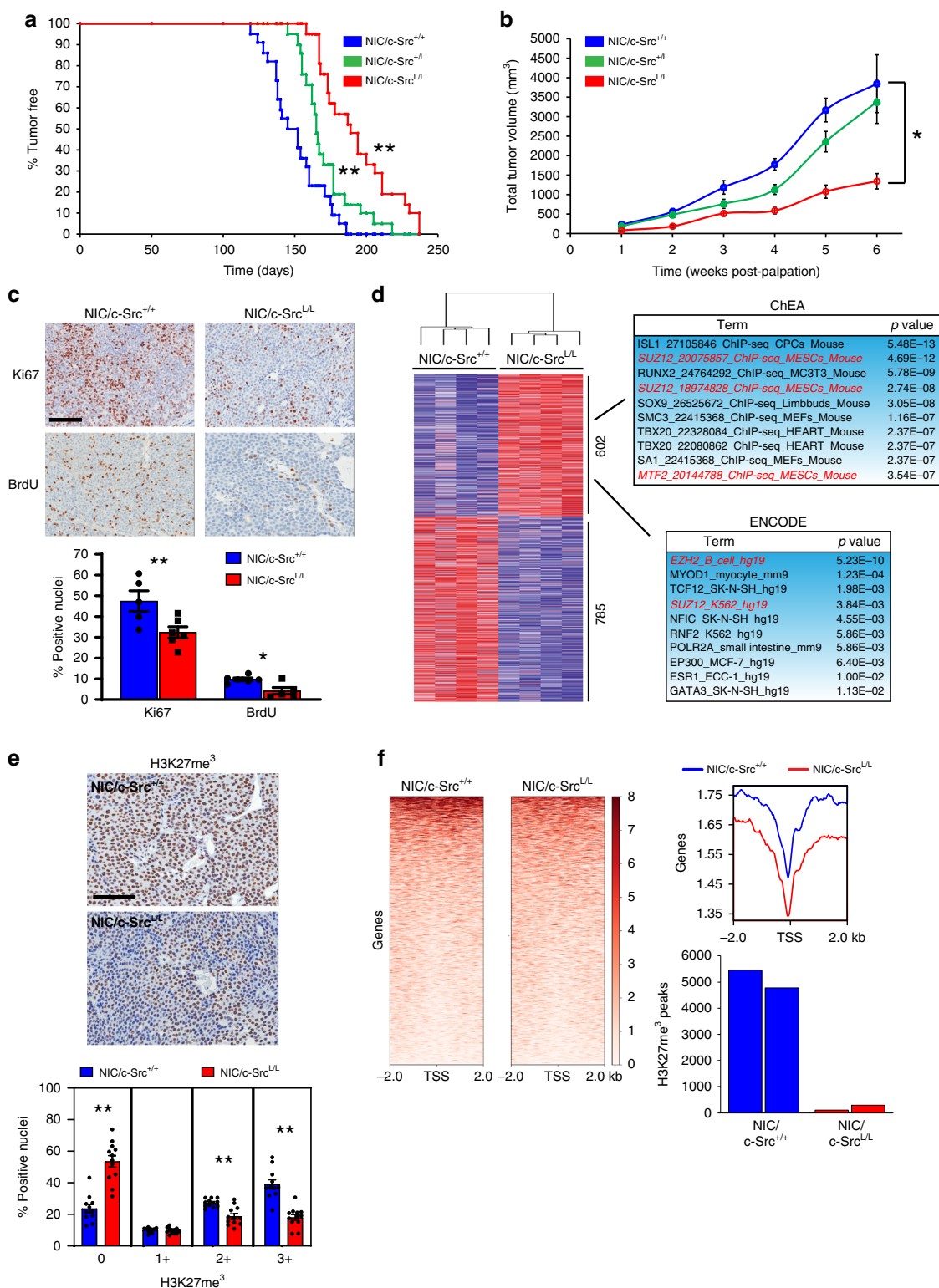

pathways as a major target of PRC2-dependent repression in this ErbB2+ model (Supplementary Fig. 2d). This is consistent with findings from the PyV mT model of luminal B breast cancer, where genes involved in neuronal differentiation and function are silenced by PRC2[21]. Interestingly, however, only 20.2% of H3K27me[3]-positive peaks in the genomes of *NIC/c-Src*[+/+] cells were mapped to genes and promoters, with the vast majority (79.8%) found in intergenic, non-coding and genomic repeat regions (Supplementary Fig. 2e). Comparison of transcripts up-regulated in c-Src-deficient cell lines with protein-coding genes associated with H3K27me[3] in control cell lines revealed an overlap of 99 genes (Supplementary Fig. 2f). To examine the relevance of this overlap, we derived a signature comprised of genes transcriptionally activated in *NIC/c-Src*[L/L] cells that exhibited H3K27me[3] within 20 kb of their transcriptional start site (TSS) in two independent *NIC/c-Src*[+/+] cell lines. We

**Fig. 1** c-Src loss impairs tumor growth and PRC2 function in ErbB2 + breast cancer. **a** Kaplan–Meier analysis of mammary tumor onset in *NIC* mice with wild-type *Src* alleles (*NIC/c-Src*$^{+/+}$, n = 26) and mice heterozygous (*NIC/c-Src*$^{+/L}$, n = 20) or homozygous (*NIC/c-Src*$^{L/L}$, n = 22) for the conditional *Src* allele. **p < 0.01; logrank test. **b** Tumor burden was determined by weekly caliper measurements. *p < 0.05; one-way ANOVA with Tukey's post-test. **c** Control and c-Src-deficient tumors were stained for Ki67 expression and incorporation of BrdU by immunohistochemistry (IHC). Top panels show representative images. Scale bar represents 100 μm. Bottom panel shows quantification of the percentage of Ki67 and BrdU-positive nuclei in tumors from five independent mice of each genotype (minimum 10,000 total nuclei, *p < 0.05, **p < 0.01; unpaired, two-tailed Student's t-test). **d** Left panel - unsupervised hierarchical clustering analysis of genes differentially up-regulated (red) and down-regulated (blue) in *NIC/c-Src*$^{L/L}$ tumors compared to *NIC/c-Src*$^{+/+}$ controls (n = 4 per genotype). Right panels - bioinformatic identification of PRC2 targets among genes up-regulated in c-Src-deficient tumors. **e** Upper two panels show representative images of H3K27me$^3$ staining of tumors from three independent *NIC/c-Src*$^{+/+}$ and *NIC/c-Src*$^{L/L}$ mice using IHC. Scale bar represents 50 μm. Bottom panel shows quantitative nuclear staining intensity scored on a scale of 0–3 using Aperio image analysis software (minimum 10000 nuclei counted per tumor, *p < 0.05, **p < 0.01; unpaired, two-tailed Student's t-test). **f** Left panels show genome-wide profiling of the H3K27me$^3$ landscape in *NIC/c-Src*$^{+/+}$ and *NIC/c-Src*$^{L/L}$ tumor cells using ChIP-Seq, with red color intensity representing tag density. Representative tag density plots (right top panel) and bar chart of the quantification of H3K27me$^3$ peaks in two independent cell lines of each genotype (right bottom panel) are shown. All error bars throughout the figure are SEM

compared the expression of the corresponding transcripts (RNA-Seq) in tumor biopsies from ERBB2+ breast cancer patients[22,23] with matched EZH2 protein expression data, revealing a strong negative correlation between EZH2 protein expression and transcript levels of PRC2 target genes from the ErbB2+ GEMM (Supplementary Fig. 2g), consistent with PRC2-mediated repression of these genes. Thus, while the de-repression of protein-coding PRC2 target genes in *NIC/c-Src*$^{L/L}$ cells is limited, a core subset of these genes may be regulated by PRC2 in human ERBB2+ breast cancer.

Overall, these findings indicate that c-Src ablation correlates with reduced PRC2 function in ErbB2-driven breast tumors. Strikingly, we observed that this correlated with a strong reduction in the protein levels of Ezh2 and Suz12 in c-Src-deficient as compared to c-Src-positive tumors (Fig. 2a, Supplementary Fig. 3a). *Ezh2* and *Suz12* mRNA levels were, however, unaffected by c-Src ablation (Supplementary Fig. 3b), thereby indicating that decreased expression of the corresponding proteins was not caused by alterations in transcription and/or mRNA stability. Proteasome inhibition moderately increased Ezh2 protein expression in c-Src-proficient, but not in c-Src-deficient cells (Supplementary Fig. 3c), indicating that increased proteosomal degradation does not play a major role in the down-regulation of PRC2 components upon c-Src ablation. We next examined whether reduced PRC2 subunit expression occurred through effects on protein synthesis. Puromycin labeling of nascent proteins[24] indicated that overall protein synthesis was decreased upon c-Src loss (Fig. 2b), while puromycin incorporation into Ezh2 protein was significantly reduced in c-Src-null cells compared to c-Src-proficient cells (Fig. 2c, Supplementary Fig. 3d). To investigate further, we used sucrose density gradient centrifugation to separate efficiently translated (associated with heavy polysomes), non-efficiently translated (associated with light polysomes) and non-translated (sub-ribosomal/mRNP fraction) transcripts[25] in extracts from c-Src-deficient and control cells. In agreement with the puromycylation assays, the number of ribosomes engaged in polysomes was decreased in c-Src-deficient cells as compared to controls (Fig. 2d). Notably, c-Src ablation shifted *Ezh2* and *Suz12* mRNAs from heavy polysomes to sub-ribosomal/mRNP fractions (Fig. 2e) without reducing their total mRNA levels (Supplementary Fig. 3e). In contrast, c-Src loss did not affect the polysome distribution of mRNAs encoding β-actin (*Actb*) or the Eed subunit of PRC2 (Fig. 2e), the expression of which was not dependent on c-Src status (Fig. 2a). These results demonstrate that c-Src loss in the context of ErbB2-driven tumorigenesis down-regulates PRC2 function by impairing translation of *Ezh2* and *Suz12* mRNAs.

**c-Src activates mTORC1 to promote *Ezh2* and *Suz12* translation.** Induction of eIF2α phosphorylation, which limits ternary complex recycling, and inhibition of mTORC1 are two major mechanisms of translational suppression[26]. Since we established that c-Src loss does not alter eIF2α phosphorylation (Supplementary Fig. 4a, b), we investigated effects on mTORC1. The expression of mTORC1 components (mTOR, Raptor, mLST8, PRAS40) and Akt-dependent phosphorylation of mTOR and PRAS40 did not differ between c-Src-deficient and control tumors (Supplementary Fig. 4c). However, mTORC1-dependent phosphorylation of Rps6 (ribosomal protein S6) and Eif4ebp1 (eukaryotic translation initiation factor 4E binding protein 1) was reduced in ErbB2-driven tumors lacking c-Src (Fig. 3a, b). Moreover, mTORC1 activation in response to amino acids, serum/growth factors and glucose was markedly attenuated in c-Src-deficient tumor cells (Supplementary Fig. 4d–f), indicating an intrinsic defect in their capacity to activate mTORC1. This was consistent with the shifted polysome profiles of c-Src-deficient cells (Fig. 2d), reflecting reduced translation of an mTORC1-dependent subset of mRNAs, as has been observed previously when mTORC1 is inhibited[27,28]. Accordingly, we observed that c-Src ablation decreased the translation of mRNAs known to be regulated by the mTORC1/4E-BP axis, including *Bcl2l1* (Bcl-xL)[29] and *Ccnd3*[27] (Supplementary Fig. 4g, h), mirroring the observed effect on *Ezh2* and *Suz12* mRNAs. Re-expression of c-Src in cells with conditional knockout of the endogenous alleles restored mTORC1 activity and Ezh2/Suz12 protein expression (Supplementary Fig. 4i). Likewise, silencing the translational suppressor Eif4ebp1, which prevents the assembly of the eIF4F complex and is inactivated via mTORC1-dependent phosphorylation[30], rescued Ezh2 protein expression in c-Src-deficient cells (Supplementary Fig. 4j). Moreover, mTOR inhibitors and silvestrol, which inhibits eIF4A, the helicase subunit of the eIF4F complex, strongly suppressed Ezh2 and Suz12 protein expression in *NIC/c-Src*$^{+/+}$ cells (Fig. 3c). Collectively, these findings demonstrate that c-Src modulates the eIF4F-dependent translation of mRNAs encoding PRC2 components via the mTORC1/4E-BP axis.

**c-Src/mTORC1 controls PRC2 expression in human breast cancer.** In agreement with previous studies linking EZH2 expression with ErbB2 signaling[31], ErbB2 kinase inhibition significantly diminished EZH2 protein expression in murine and human cells in vitro, while ErbB2-driven pre-malignant mammary lesions in vivo up-regulated Ezh2 protein expression in a c-Src-dependent manner (Supplementary Fig. 5a, b). To examine potential relationships between c-SRC activity, mTORC1 activation and PRC2 expression in human breast cancer, we analyzed a tissue microarray (TMA) series containing primary samples from a cohort of 292 patients, revealing that markers of SFK and mTORC1 activation correlate positively with each other and with EZH2 protein levels (Fig. 4a, b). We used a second, independent

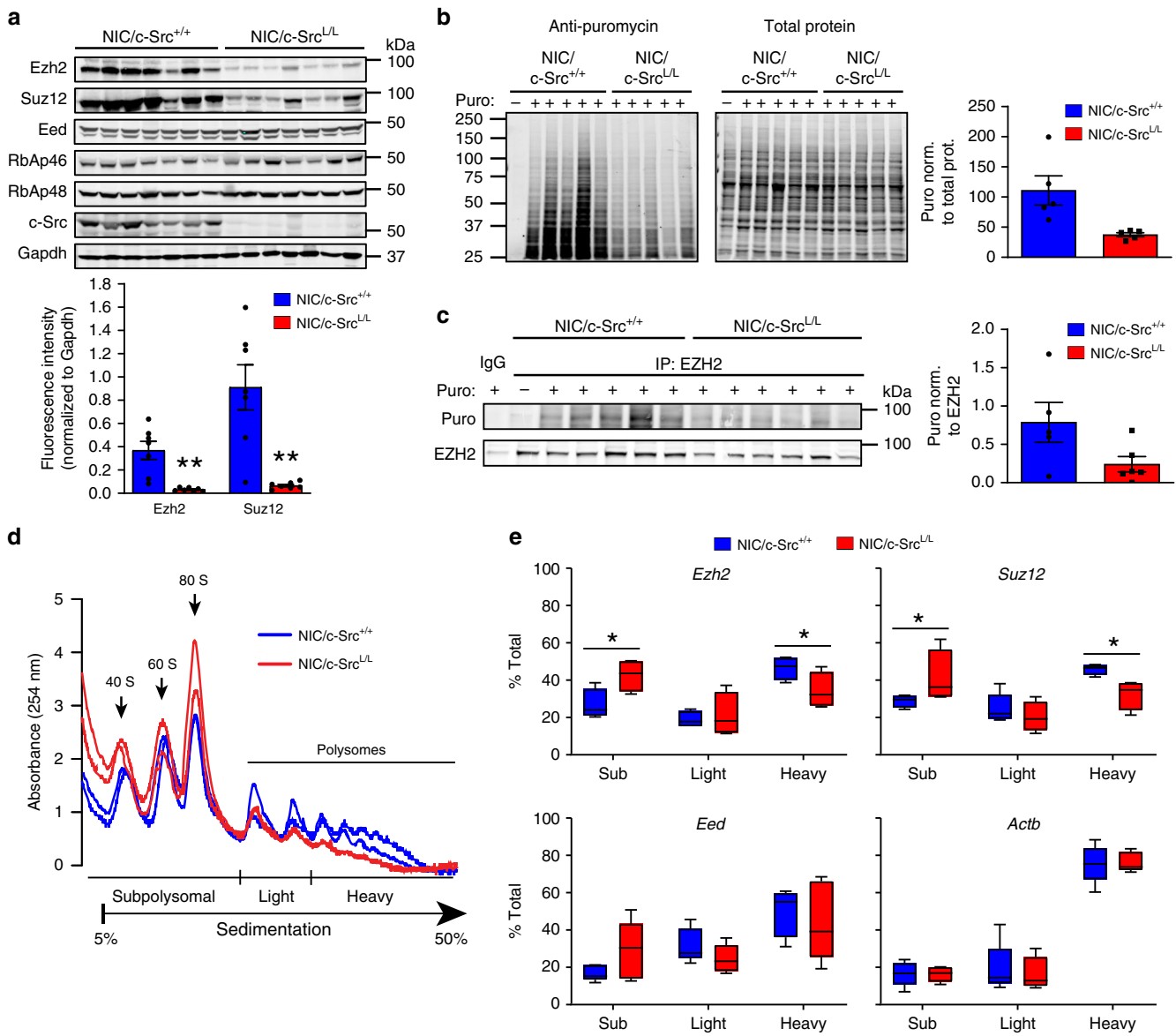

**Fig. 2** c-Src is required for efficient translation of PRC2 component mRNAs. **a** Immunoblot of PRC2 components in control and c-Src-deficient tumors. Bar chart shows quantification of fluorescent immunoblot data normalized to the loading control (Gapdh). **b** Protein synthesis in *NIC/c-Src$^{+/+}$* and *NIC/c-Src$^{L/L}$* cells was assessed using puromycin incorporation with immunoblotting (Surface sensing of translation (SUnSET) assay). Images show detection of puromycylated nascent proteins (left) and total cellular protein (right) following a 30 min. treatment with puromycin (puro). Bar chart shows the puromycin signal in each lane normalized to total protein levels ($n = 5$ per genotype). **c** Ezh2 was immunoprecipitated from *NIC/c-Src$^{+/+}$* and *NIC/c-Src$^{L/L}$* cells treated as in **b** and newly synthesized Ezh2 was detected by immunoblotting with anti-puromycin (puro) and anti-Ezh2 antibodies. Immunoprecipitation with non-specific IgG and Ezh2 immunoprecipitation from untreated cells were used as controls. Bar chart shows quantification of the puromycin signal in Ezh2 immunoprecipitations normalized to the total Ezh2 levels precipitated ($n = 5$ per genotype). **d** Representative polysome profiles of two *NIC/c-Src$^{+/+}$* and two *NIC/c-Src$^{L/L}$* tumor cell lines. Profiles are representative of five biological replicates per genotype. **e** qRT-PCR analysis of mRNAs isolated from polysome profiling of *NIC/c-Src$^{+/+}$* and *NIC/c-Src$^{L/L}$* tumor cell lines (5 biological replicates per genotype). In the box and whisker plots, the center line represents the median, boundaries are the 25th and 75th percentiles and whiskers were calculated according to Tukey's method. For all data, * = $p < 0.05$ and ** = $p < 0.01$ (unpaired, two-tailed Student's *t*-test). For all bar charts, error bars are SEM

TMA series containing 131 patient samples of ERBB2+ breast cancer to confirm these correlations specifically within this subtype (Fig. 4c). Functional relationships were assessed in a human breast cancer setting by treating immunocompromised mice bearing patient-derived xenografts (PDXs) established from two ERBB2+ tumors with the SFK inhibitor Dasatinib or the active site-directed mTOR inhibitor AZD2014, both of which are currently involved in multiple clinical trials. These inhibitors attenuated tumor growth, with AZD2014 exerting the most significant effects (Supplementary Fig. 5c, d) and also reduced

mTORC1 activation (as assessed by EIF4EBP1 phosphorylation) and the expression of EZH2 and SUZ12 proteins (Fig. 4d, e). We confirmed that these effects were not due to changes in the expression of mTORC1 component proteins (Supplementary Fig. 5e), although phosphorylation of mTOR (Ser2448) and PRAS40 (Thr246) was reduced in Dasatinib- and AZD2014-treated tumors. Reduced EZH2 expression and diminished H3K27me$^3$ in Dasatinib- and AZD2014-treated tumors were validated by immunostaining (Fig. 4f, Supplementary Fig. 5f). Allosteric and active site-directed mTOR inhibitors also reduced

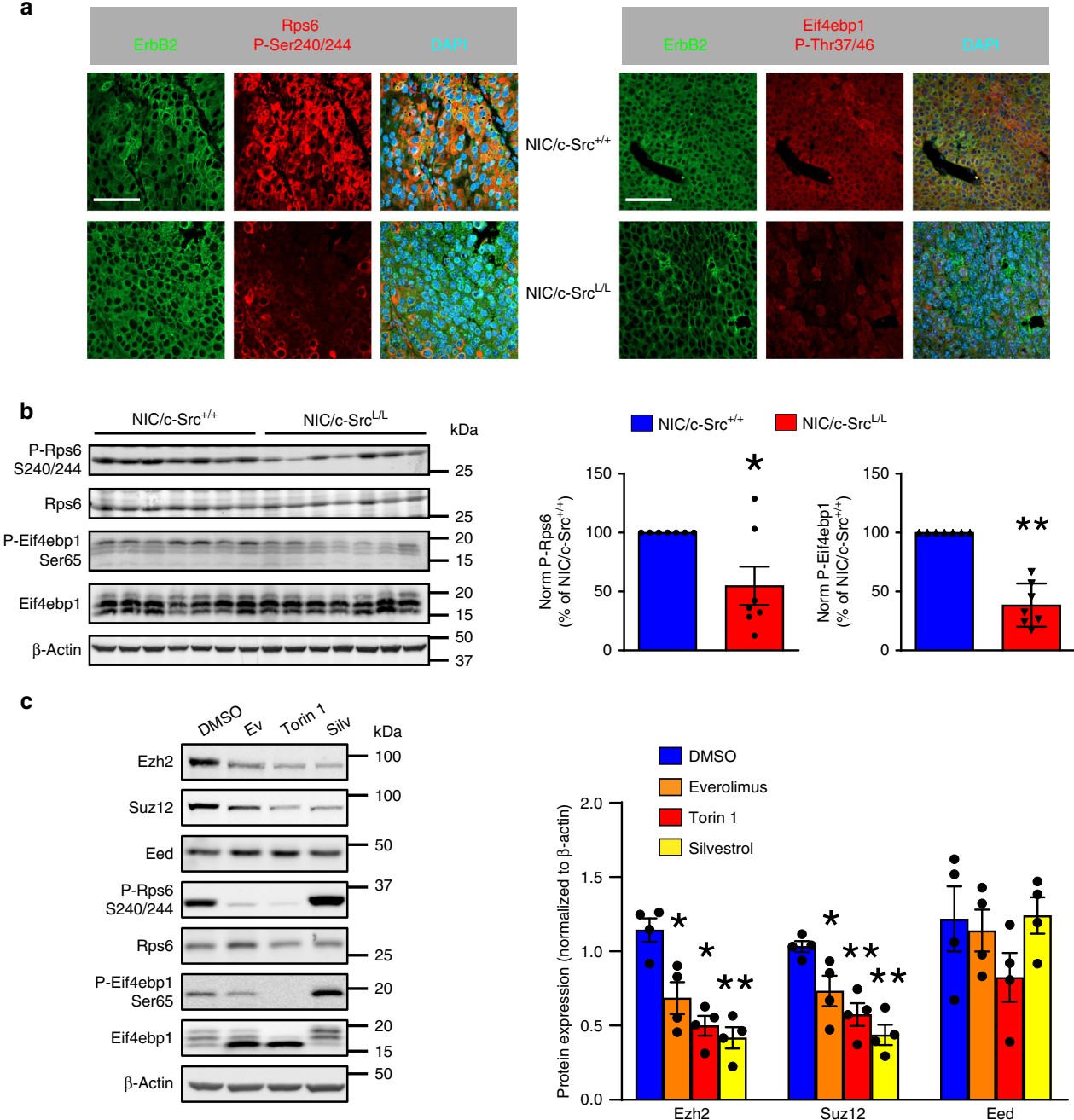

**Fig. 3** mTORC1 activity is impaired in ErbB2 + breast cancer cells lacking c-Src. **a** IF analysis of ErbB2 and phosphorylated forms of Rps6 and Eif4ebp1 in control and c-Src-deficient ErbB2 + mammary tumors. Scale bars represent 50 μm. **b** Left panel – immunoblots of Rps6 and Eif4ebp1 phosphorylation in control and c-Src-deficient ErbB2 + mammary tumors. Right panel - bar charts show quantification of phosphorylated Rps6 and Eif4ebp1 proteins ($n = 7$ tumors per genotype). **c** Left - representative immunoblots of PRC2 expression and mTORC1 signaling in *NIC/c-Src*$^{+/+}$ ErbB2 + breast cancer cells treated with the indicated inhibitors for 24 h (Ev: everolimus, 10 nM; Torin 1, 250 nM; Silv: silvestrol, 25 nM). Right - Quantification of Ezh2, Suz12 and Eed protein levels in four independent cell lines following the indicated treatments. All data in this figure are presented as mean ± SEM, where $* = p < 0.05$ and $** = p < 0.01$; unpaired, two-tailed Student's $t$-test

EZH2 and SUZ12 protein expression in human ERBB2+ breast cancer cell lines (SkBr3 and MDA-MB-361), but not a triple-negative breast cancer (TNBC) cell line (MDA-MB-231), in a manner commensurate with their ability to decrease EIF4EBP1 phosphorylation (Supplementary Fig. 5g). A similar decrease in EZH2 and SUZ12 expression was observed in ERBB2+, but not TNBC, cell lines treated with silvestrol or with two independent SFK inhibitors, Dasatinib and eCF506, which exhibits greatly enhanced specificity for the Src family[32] (Supplementary Fig. 5g).

Finally, we examined EZH2 protein and mRNA expression and EIF4EBP1 phosphorylation in an ERBB2+ patient cohort for which both tissue sections and RNA-Seq data were available (as in Supplementary Fig. 2g)[22,23]. Strikingly, we found that EZH2 protein expression did not significantly correlate with *EZH2* mRNA expression, but strongly correlated with phosphorylation of EIF4EBP1, and hence activity of mTORC1 (Fig. 4g). This is consistent with the pivotal role of the mTORC1/4E-BP axis in regulating EZH2 expression in ERBB2+ breast cancer.

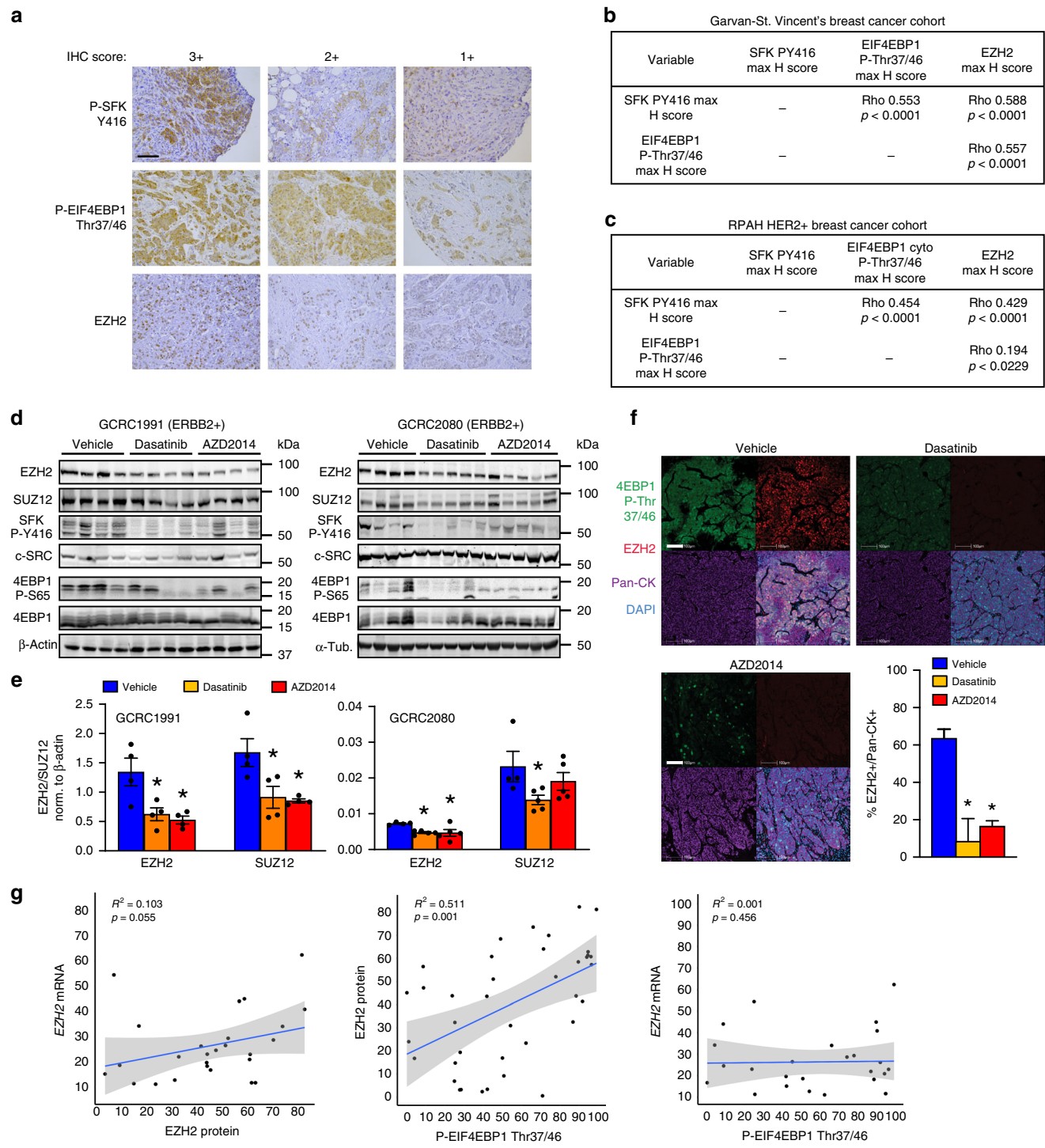

**c-Src Drives PRC2 overexpression by regulating bioenergetics.** c-Src can cooperate with RTKs to activate Ras-MAPK and PI3K-Akt pathways that act upstream of mTORC1[15,33,34]. However, phosphorylation of ErbB2, Egfr, ErbB3, Erk, and Akt was unaffected by c-Src ablation in ErbB2+ tumors, as was Akt-dependent phosphorylation of Tuberous Sclerosis Complex 2 (Tsc2) (Supplementary Fig. 6a). Thus, in ErbB2-driven breast cancer, c-Src appears to activate mTORC1 via a mechanism independent of canonical RTK-driven pathways. Because mTORC1 is modulated by metabolic inputs[35] and c-Src is implicated in metabolic regulation[36,37], we examined known metabolic regulators of mTORC1, including energy and amino acid levels. Strikingly,

phosphorylation of the central energy sensor 5′ adenosine monophosphate-activated protein kinase (AMPK) and of its substrate acetyl-CoA carboxylase (ACC) were elevated in c-Src-deficient, as compared to c-Src-positive, tumor tissue and cell lines (Fig. 5a, b), correlating with an increased AMP:ATP ratio in c-Src-deficient cells (Fig. 5c, Supplementary Fig. 6b). c-Src-deficient cells also exhibited significant reductions in the steady-state levels of most amino acids (Supplementary Fig. 6c). Although this was insufficient to activate an amino acid starvation response via GCN2-dependent eIF2α phosphorylation (Supplementary Fig. 4a, b), it could potentially impair mTORC1 activation. However, AMPK-dependent phosphorylation of

**Fig. 4** c-Src and mTORC1 drive EZH2 protein expression in human breast cancer. **a** Tissue microarrays (TMAs) containing tumor samples from breast cancer patients were stained with antibodies against EZH2, phosphorylated Src family kinases (SFK) and phosphorylated EIF4EBP1. Example images indicate scoring by a specialist breast cancer pathologist (S.A.O.) with decreased scoring from left to right. Scale bar indicates 100 μm. **b**, **c** Associations between staining intensities for each marker in **a** on two breast cancer TMAs. The Garvan-St. Vincent's cohort **b** contained 292 samples from patients of all subtypes, while the Royal Prince Albert Hospital (RPAH) cohort **c** contained samples from 131 HER2 + breast cancer patients. Rho and p values were calculated using Spearman's rank-order correlation analysis. **d**, **e** ERBB2 + breast cancer patient-derived xenografts GCRC1991 and GCRC2080 were orthotopically implanted into NOD/SCID/gamma (NSG) mice which were treated with the SFK inhibitor Dasatinib ($n = 7$), the ATP-competitive mTOR kinase inhibitor AZD2014 ($n = 7$) or vehicle ($n = 6$). EZH2 and SUZ12 levels and the activity of SFK and mTORC1 were determined by immunoblotting of tumor tissue lysates **d**. Bar charts **e** show quantification of EZH2 and SUZ12 levels normalized to the loading control (ACTB) using fluorescent immunoblotting ($n = 4$, *$p < 0.05$, unpaired Student's t-test vs vehicle). **f** Immunofluorescence analysis of EZH2 expression and phosphorylation of EIF4EBP1 in ERBB2 + PDX tumors following administration of inhibitors or vehicle control. Bottom right panel - digital pathology analysis software (HALO) was used to quantify EZH2 staining in tumor cells (positive for human-specific pan-cytokeratin staining). Scale bars indicate 100 μm (minimum 10000 cells counted per tumor, *$p < 0.05$, unpaired, two-tailed Student's t-test). **g** EZH2 protein expression and EIF4EBP1 phosphorylation were quantified in samples from ERBB2 + breast cancer patients using immunofluorescence with digital pathology analysis as in **f**. Pearson's correlation analysis was used to correlate these parameters with each other and with *EZH2* mRNA levels (RNA-Seq reads) determined in the same samples. Left panel shows the correlation between *EZH2* mRNA and protein, middle panel shows the correlation between EZH2 protein and phospho-EIF4EBP1, right panel shows the correlation between *EZH2* mRNA and phospho-EIF4EBP1. All error bars in this figure are SEM

Raptor[35] and Tsc2[38], two important mediators of mTORC1 suppression by energy stress, was also elevated in c-Src-deficient tumors (Supplementary Fig. 7a). This led us to characterize the bioenergetic state of c-Src-deficient tumor cells in more detail. c-Src ablation significantly decreased the basal, ATP synthesis-coupled and maximal oxygen consumption rates (OCR) of ErbB2+ tumor cells, indicating an overall suppression of respiration (Fig. 5d, e). Because mTORC1 promotes the translation of a subset of mRNAs encoding mitochondrial proteins, leading to impaired mitochondrial function in cells where mTOR is inhibited[39], we examined the levels of known mTORC1-dependent mitochondrial proteins (Atp5o, Tfam) and components of the electron transport chain (ETC) in control and c-Src-deficient cells. Consistent with previous findings[39], we observed reduced expression of Atp5o and Tfam (Supplementary Fig. 7b) and diminished levels of some ETC components including Ndufb8 (complex I) and CoxIV (complex IV) in cells lacking c-Src (Supplementary Fig. 7c). In contrast, no difference was observed in the mitochondrial DNA to nuclear DNA (mtDNA: nDNA) ratio, suggesting that alterations in mitochondrial protein expression are not due to reduced mitochondrial biogenesis (Supplementary Fig. 7d). In addition to effects on OXPHOS, c-Src deficiency increased the extracellular acidification rate (ECAR) (Fig. 5e), which was corroborated by elevated glucose consumption and lactate excretion (Fig. 5f). These findings argue that c-Src deficiency reduces OXPHOS and elevates glycolysis, a metabolic shift reflected in the gene expression profiles of c-Src-deficient tumors, which were enriched in signatures of glycolysis (Fig. 5g).

Changes in OCR and ECAR induced by c-Src deletion strongly resemble those caused by biguanides[40], which inhibit complex I of the ETC. This reduces ATP synthesis and increases the AMP/ATP ratio[41,42], inactivating mTORC1 through both AMPK-dependent and AMPK-independent mechanisms[43,44]. Linking this metabolic remodeling to PRC2 component mRNA translation, analysis of human breast cancer cell translatomes[27] showed that the biguanide metformin, similarly to allosteric (rapamycin) or active site-directed (PP242) mTOR inhibitors, reduced the translational efficiency of *EZH2* mRNA (Supplementary Fig. 7e). Treatment of ErbB2-driven breast cancer cells with the biguanide phenformin, which directly inhibits OXPHOS regardless of the presence of c-Src, activated AMPK and inhibited mTORC1, markedly reducing Ezh2 and Suz12 protein expression (Fig. 5h, i). Multiple compounds targeting OXPHOS, including the unrelated complex I inhibitor, rotenone, as well as the mitochondrial ATP synthase inhibitor oligomycin A and the uncoupling agent CCCP (Fig. 5h, i) replicated these effects. Moreover, direct allosteric

activation of AMPK by GSK621 and PF-06409577 also suppressed Ezh2 and Suz12 protein expression concomitantly with mTORC1 inhibition (Fig. 5h, i). None of these treatments affected the expression of mTORC1 component proteins (Supplementary Fig. 7f), although Akt-dependent phosphorylation of PRAS40 was reduced by inhibition of complex I or ATP synthase. Collectively, these results demonstrate that, in the context of ErbB2-driven breast cancer, c-Src loss reprograms bioenergetics and induces energy stress, reducing mTORC1 activity and thereby down-regulating the translation of PRC2 components.

To determine whether the metabolic phenotypes of chronically c-Src-deficient ErbB2+ tumor cells are reproduced by acute c-Src loss, we expressed Cre recombinase using adenoviral vectors in ErbB2+ breast cancer cells with homozygous conditional Src alleles. The loss of c-Src protein was complete at 96 h post-infection (Supplementary Fig. 8a), correlating with an increased AMP:ATP ratio, AMPK activation, mTORC1 suppression, and reduced Ezh2 and Suz12 protein expression (Fig. 6a, b and Supplementary Fig. 8b). Acute c-Src loss (Fig. 6c, d) or inhibition of SFKs using Dasatinib or eCF506 (Supplementary Fig. 8c, d) significantly suppressed OXPHOS, including basal respiration and respiration coupled to mitochondrial ATP synthesis. However, in contrast to cells from c-Src-deficient transgenic tumors, extracellular acidification, glucose consumption and lactate excretion were also suppressed by acute c-Src ablation (Fig. 6d, e). c-Src re-expression in c-Src-null cells elevated OCR to control levels, indicating restoration of OXPHOS, which was dependent on kinase activity since the effects were reversed by SFK inhibitors (Supplementary Fig. 8d). SFK inhibitors did not affect OCR in cells lacking c-Src expression, suggesting that their effects were due to inhibition of c-Src (Supplementary Fig. 8d). These observations confirm that c-Src supports OXPHOS capacity in ErbB2+ breast cancer cells, while arguing that metabolic reprogramming towards enhanced glycolysis is an adaptation that occurs during tumor evolution in the absence of c-Src. To examine the role of AMPK in these phenotypes, we genetically depleted its catalytic subunits (AMPKα1/α2). While this partially restored OCR, it returned ECAR, glucose consumption and lactate excretion to the levels seen in c-Src-proficient controls and rescued both mTORC1 signaling and expression of Ezh2 and Suz12 proteins (Fig. 7a–e).

In addition to AMP binding to the AMPKγ subunit, phosphorylation of Thr172 in the activation loop of the AMPKα subunit, performed by LKB1 during energy stress and by CAMKKβ in response to Ca2+ influx, is an important event in

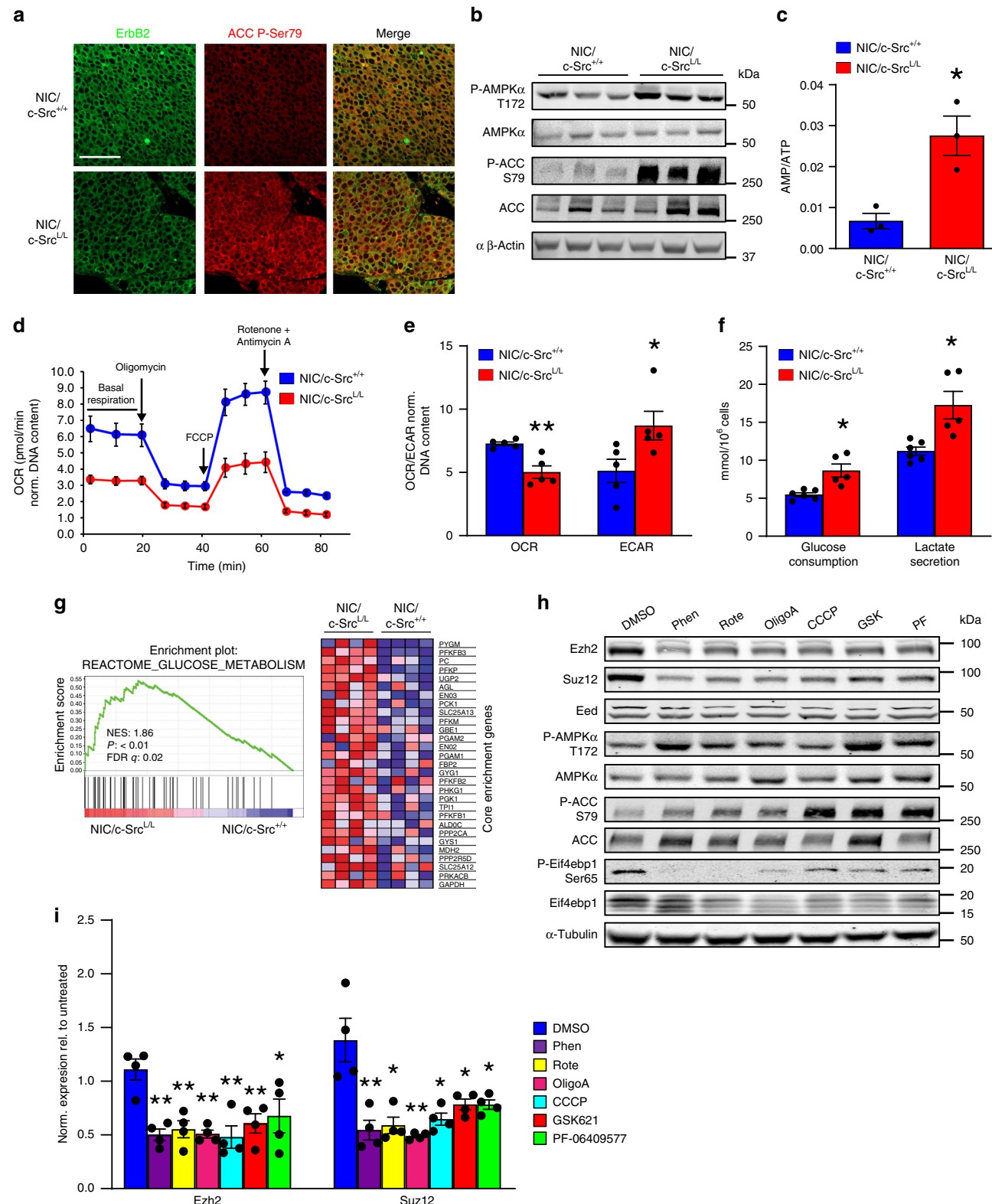

AMPK activation[45]. Notably, CAMKKβ was recently implicated in AMPK activation during amino acid starvation[45,46]. To examine the contributions of these regulators to AMPK activation in *NIC/c-Src*[+/+] and *NIC/c-Src*[L/L] cells, we used the specific Camkkβ inhibitor STO-609[47] (Supplementary Fig. 9a) and silenced Lkb1 expression with three independent siRNAs (Supplementary Fig. 9b). While Camkkβ inhibition mainly

affected AMPK activity in *NIC/c-Src*[+/+] cells, silencing Lkb1 reduced the phosphorylation of AMPK and ACC in both *NIC/c-Src*[+/+] and *NIC/c-Src*[L/L] cells. These findings are in agreement with a previous study using the same model, where Lkb1 deletion abolished the majority of AMPK activity[40]. However, these results may also be consistent with Lkb1 and Camkkβ acting in parallel or cooperatively during AMPK activation[48].

**Fig. 5** Impaired OXPHOS and induction of energy stress reduce PRC2 expression. **a** IF analysis of ErbB2 expression and ACC phosphorylation in c-Src-proficient and -deficient ErbB2 + mammary tumors. Representative images from five tumors of each genotype. Scale bar represents 50 μm. **b** Representative immunoblot showing AMPK activation and ACC phosphorylation in *NIC/c-Src*$^{+/+}$ and *NIC/c-Src*$^{L/L}$ cell lines. **c** LC/MS analysis of AMP: ATP ratios in *NIC/c-Src*$^{+/+}$ and *NIC/c-Src*$^{L/L}$ cell lines (3 cell lines per genotype). **d** Basal, maximal (FCCP), ATP-synthesis-coupled (Oligomycin A), and non-mitochondrial (rotenone/antimycin A) oxygen consumption rates (OCRs) of *NIC/c-Src*$^{+/+}$ and *NIC/c-Src*$^{L/L}$ cells. Representative of 5 cell lines per genotype. **e** Quantification of basal OCR and extracellular acidification rates (ECAR) of *NIC/c-Src*$^{+/+}$ and *NIC/c-Src*$^{L/L}$ cell lines. (5 cell lines per genotype). **f** Glucose consumption and lactate levels in conditioned media from *NIC/c-Src*$^{+/+}$ and *NIC/c-Src*$^{L/L}$ cell lines (5 cell lines per genotype). **g** Representative plot and heatmap from GSEA showing increased expression of genes associated with glucose metabolism in *NIC/c-Src*$^{L/L}$ tumors relative to *NIC/c-Src*$^{+/+}$ controls (4 tumors per genotype). **h** *NIC* cells were treated for 24 h with the mitochondria-targeting agents phenformin (Phen, 100 μM), rotenone (Rote, 0.5 μM), Oligomycin A (OligoA, 1 μM) or CCCP (1 μM) or the allosteric AMPK activators GSK621 (GSK, 25 μM) or PF-06409577 (PF, 50 μM) for 24 h. Representative immunoblots of PRC2 component expression, AMPK and mTORC1 pathway activation are shown. **i** Quantification of Ezh2 and Suz12 protein expression in four independent cell lines treated as in **h**. All bar charts show mean±SEM, with * = *p* < 0.05 and ** = *p* < 0.01 (unpaired, two-tailed Student's *t*-test)

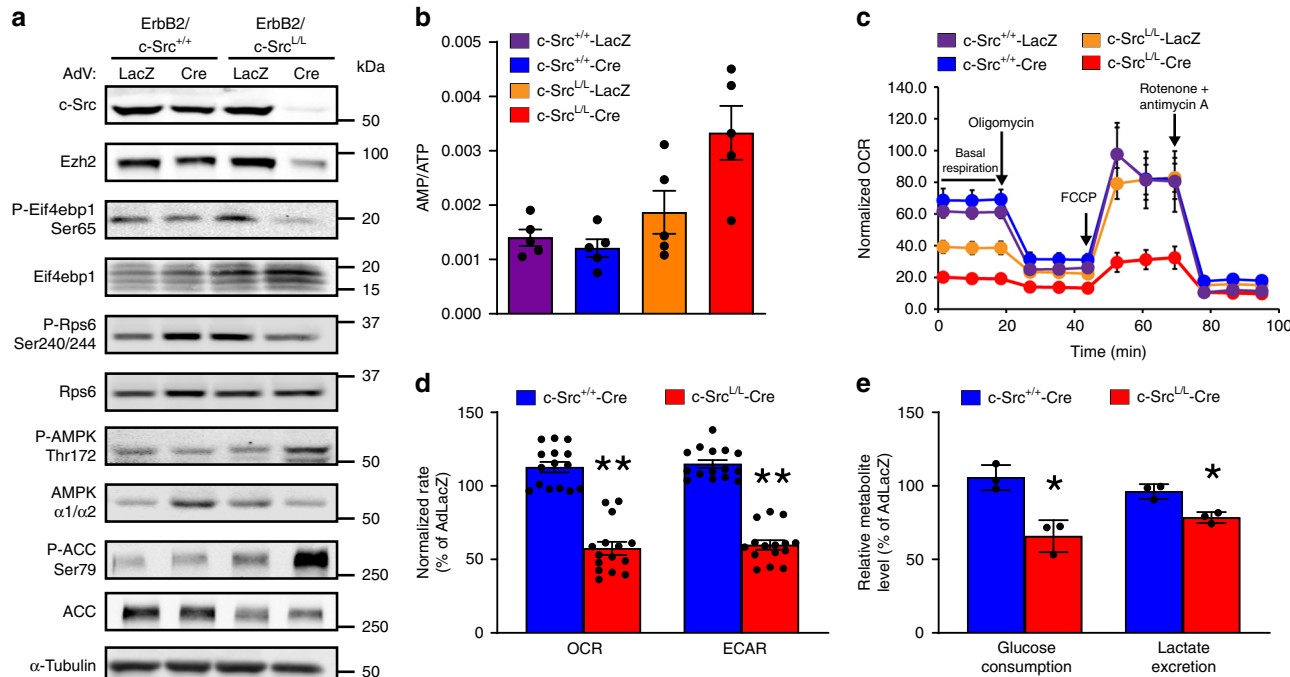

**Fig. 6** Acute c-Src ablation diminishes both OXPHOS and glycolysis. **a** ErbB2 + murine breast cancer cells with wild-type (*c-Src*$^{+/+}$) or conditional (*c-Src*$^{L/L}$) *Src* alleles were infected with adenoviruses bearing Cre recombinase or LacZ. Immunoblots show c-Src expression, mTORC1 and AMPK signaling and PRC2 component expression at 96 h post-infection. Data are representative of 3 cell lines per genotype. **b** LC/MS analysis of AMP:ATP ratios in control (LacZ) and Cre-expressing cell lines (*n* = 3 cell lines per genotype). **c** Basal, maximal (FCCP), ATP-synthesis-coupled (Oligomycin A), and non-mitochondrial (rotenone/antimycin A) oxygen consumption rates (OCRs) of ErbB2-expressing *c-Src*$^{+/+}$ and *c-Src*$^{L/L}$ breast cancer cell lines (n = 2 per genotype) infected with Cre or LacZ adenoviruses. Representative of 2 cell lines per genotype. **d** Basal OCR and ECAR of *c-Src*$^{+/+}$ and *c-Src*$^{L/L}$ cell lines infected with Cre adenoviruses, expressed as a percentage of the rate measured in the LacZ control for each cell line. **e** Glucose consumption and lactate levels in conditioned media from *c-Src*$^{+/+}$ and *c-Src*$^{L/L}$ cell lines infected with Cre adenoviruses, expressed as a percentage of levels observed in LacZ-expressing controls. All bar charts show mean ±SEM, with * = *p* < 0.05 and ** = *p* < 0.01 (unpaired, two-tailed Student's *t*-test)

**Ezh2 is required for ErbB2-driven mammary tumorigenesis.** To directly address the requirement for PRC2/Ezh2 in ErbB2-driven breast cancer progression, we performed a series of experiments where Ezh2 was ablated or down-regulated in ErbB2+ breast cancer models. Remarkably, conditional deletion of *Ezh2* (*Ezh2*$^{L/L}$)[2] in an ErbB2-driven GEMM induced a near-complete block in mammary tumorigenesis, with focal tumors arising after a long latency and with significantly reduced penetrance after over a year of follow-up (Fig. 8a). Ezh2 plays a complex role during mammary development, with studies showing that it regulates mammary epithelial stem cell populations[49,50] and that its deletion accelerates[51] or delays[49,50] pubertal ductal outgrowth. Using *MMTV-Cre/Ezh2*$^{L/L}$ mice and *MMTV-Cre/Ezh2*$^{+/+}$ controls on a pure FVB/N genetic background, we determined that mammary epithelial *Ezh2* ablation

transiently delayed outgrowth of the mammary ductal tree, in agreement with previous studies[49]. *MMTV-Cre/Ezh2*$^{L/L}$ mice exhibited normal ductal architecture and no significant difference in outgrowth by 12 weeks of age (Supplementary Fig. 10a). Whole-mounted mammary glands from 16-week-old *NIC/Ezh2*$^{L/L}$ mice contained a complete ductal tree with a normal gross histological appearance, while age-matched control *NIC/Ezh2*$^{+/+}$ mice manifested visible mammary epithelial lesions (Supplementary Fig. 10b). ErbB2-positive/Ezh2-null epithelial cells in normal ducts and small hyperplastic lesions in mammary glands from *NIC/Ezh2*$^{L/L}$ mice (Fig. 8b) were largely Ki67-negative, whereas prominent Ki67 staining was observed in ErbB2-positive/Ezh2-positive cells in age-matched control *NIC/Ezh2*$^{+/+}$ mammary glands (Fig. 8c, d). These findings suggest that, while Ezh2-deficient mammary epithelial cells can be transformed by

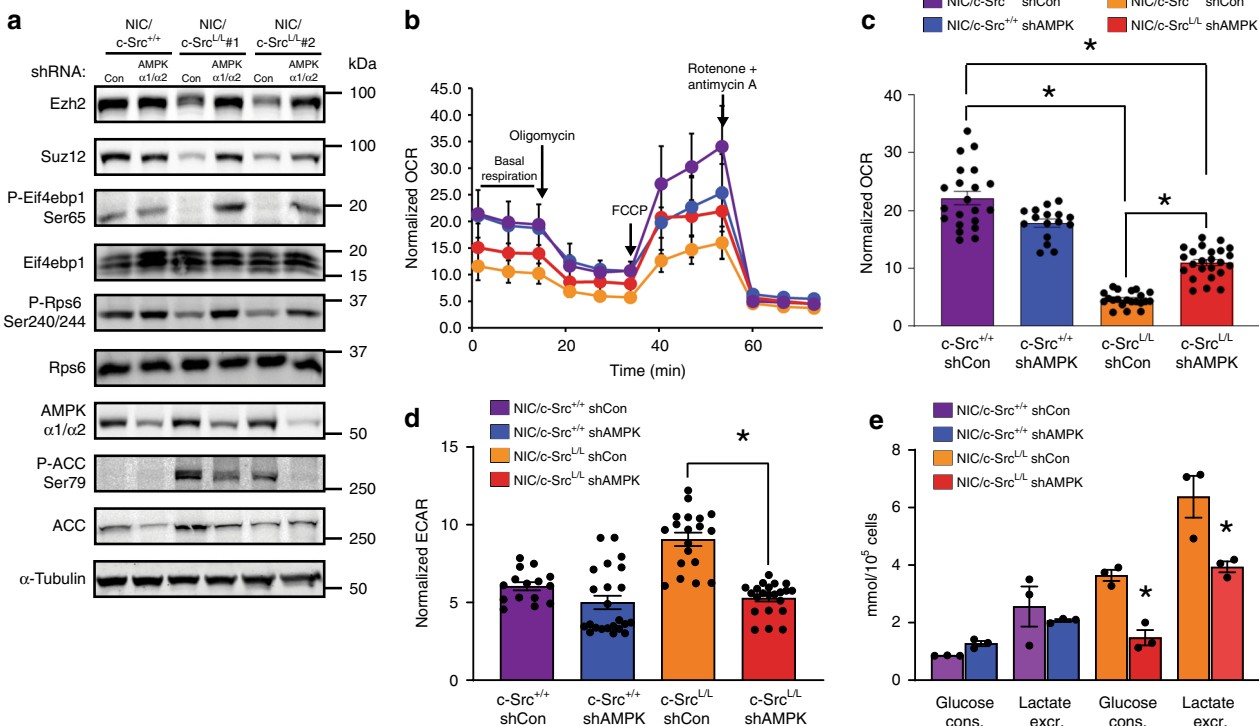

**Fig. 7** AMPKα silencing partially reverses metabolic phenotypes of c-Src-deficient cells. **a** NIC/c-Src[+/+] cells and two independent NIC/c-Src[L/L] cell lines were transduced with a retrovirus bearing shRNAs against Prkaa1 and Prkaa2 (AMPKα1 and AMPKα2). Immunoblots show expression of PRC2 components, mTORC1 activation and AMPK α1/α2 expression and activity. **b** Basal, maximal (FCCP), ATP-synthesis-coupled (Oligomycin A), and non-mitochondrial (rotenone/antimycin A) oxygen consumption rates (OCRs) of NIC cell lines expressing control or AMPK α1/α2 shRNAs. Representative of 2 cell lines per genotype. **c, d** Quantification of basal OCR and ECAR of cells as in **b**. **e** Glucose consumption and lactate levels in conditioned media from cells as in **b**. All bar charts show mean ± SEM, with * = $p < 0.05$ and ** = $p < 0.01$ (unpaired, two-tailed Student's t-test)

ErbB2, tumor progression in the absence of Ezh2 is blocked at an early hyperplastic stage. To examine the role of Ezh2 methyltransferase activity in ErbB2-driven mammary tumor progression, we treated 10 week-old MMTV-NIC mice with the Ezh2 inhibitor GSK126, which competes with S-adenosylmethionine (SAM) for binding to the EZH2 SET domain[52]. After 10 weeks of treatment, a largely normal ductal tree was observed in mice treated with GSK126, with evidence of small hyperplastic lesions, morphologically similar to those observed in the Ezh2 genetic knockout, which demonstrated loss of H3K27me[3] (Fig. 8e and Supplementary Fig. 10c). In contrast, the mammary glands of vehicle-treated mice had undergone widespread transformation, with abundant hyperplastic and adenomatous lesions exhibiting H3K27me[3] (Fig. 8e and Supplementary Fig. 10c). As with Ezh2 genetic ablation, this phenotype correlated with significantly reduced Ki67 expression in the ErbB2-positive mammary epithelial cells of mice treated with the Ezh2 inhibitor (Fig. 8f).

These data indicate that Ezh2 has pivotal functions in ErbB2-driven mammary tumorigenesis that are unlikely to be related to confounding effects on mammary gland development. To test the effect of downregulating Ezh2 expression, rather than complete loss of function through conditional gene targeting, and to determine effects on the growth of established tumor cells, we stably silenced Ezh2 in cells from the MMTV-NIC model. This significantly reduced proliferation in vitro and impaired orthotopic tumor growth in vivo (Fig. 9a, b). Two structurally unrelated Ezh2 inhibitors, GSK126 and EPZ6438[53], also significantly impaired the proliferation of ErbB2-driven breast cancer cells in vitro (Fig. 9c). In contrast to NIC/c-Src[+/+] cells, the growth of NIC/c-Src[L/L] cells, which have reduced Ezh2 and Suz12 expression and diminished PRC2 activity, was not

significantly affected by GSK126 (Supplementary Fig. 10d). Collectively, these data confirm that Ezh2 overexpression and catalytic activity strongly promote the proliferation of ErbB2-driven breast cancer cells.

## Discussion

Persistent activation of mTORC1, a master coordinator of cellular metabolism and growth, reprograms the translatome to support the malignant state[28,54]. By enhancing eIF4F-dependent translation, mTORC1 activation increases the synthesis of mitochondrial proteins[39] and transcription factors promoting mitochondrial biogenesis[54,55], thereby stimulating OXPHOS. mTORC1 also increases glycolytic flux[56,57] and drives the translation of mRNAs encoding transcription factors that up-regulate glycolysis, including HIF-1α[58]. Here, we show that mTORC1 activation shapes the transcriptome at the epigenetic level by promoting the translation of PRC2 components, establishing an epigenetic landscape underpinning the progression of ErbB2-driven mammary tumors (Fig. 9d). These findings place PRC2 components within the spectrum of mRNAs up-regulated by mTORC1 to drive unrestrained cell growth and proliferation.

We found that mTORC1 activation in ErbB2+ breast cancer cells was exquisitely sensitive to c-Src activity through a tumor cell-intrinsic mechanism involving bioenergetic reprogramming, rather than modulation of growth factor-dependent pathways such as PI-3K/Akt[33,34] or changes in the expression of mTORC1 subunits. Dasatinib-induced changes in the phosphorylation of mTORC1 subunits (mTOR Ser2448 and PRAS40 Thr246) in PDX models (Supplementary Fig. 9e) may be due to inhibition of kinases other than c-Src that act upstream of PI-3K/Akt, as these effects are not observed in systems where c-Src is

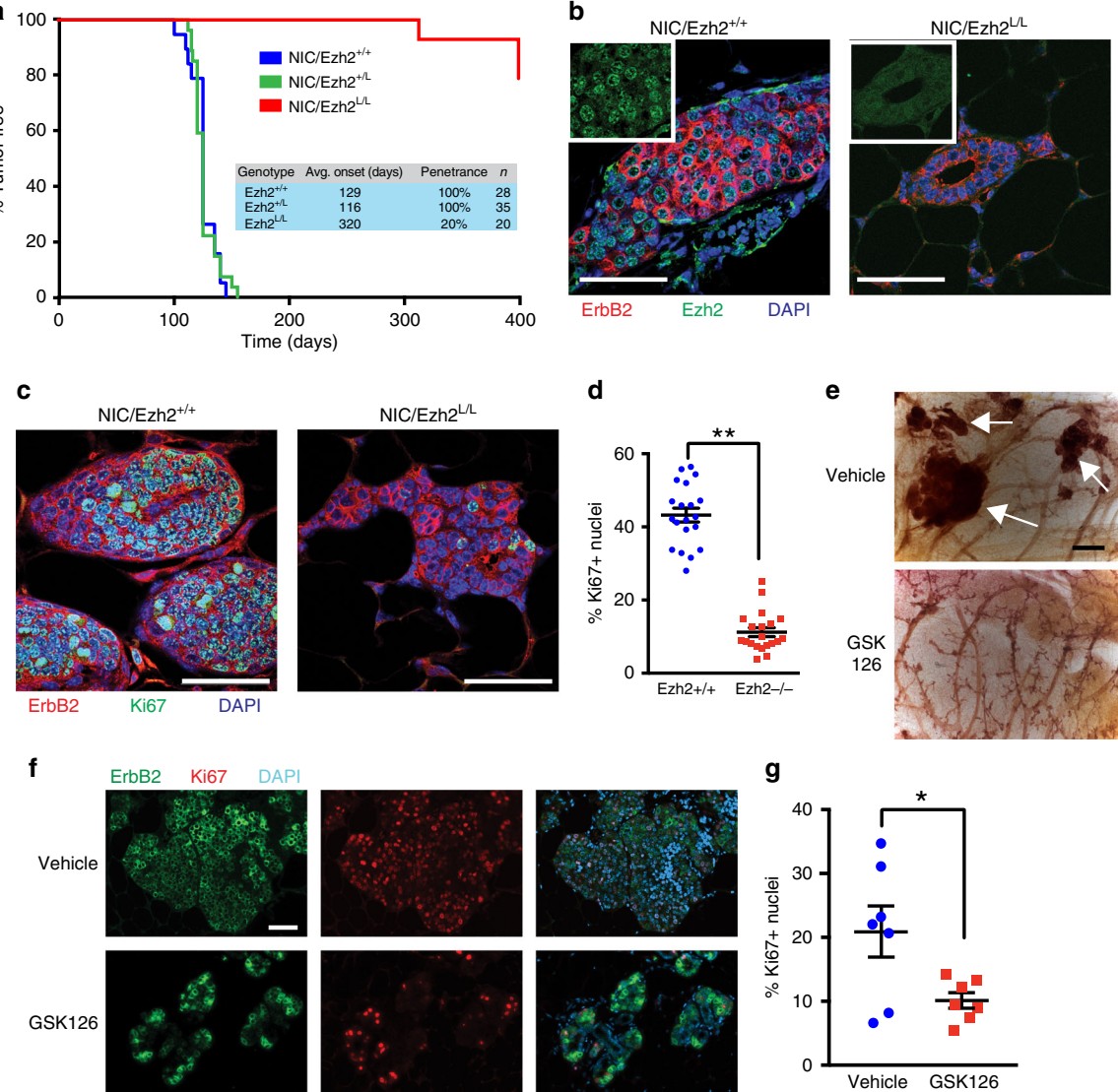

**Fig. 8** Ezh2 is required for ErbB2-driven mammary epithelial transformation. **a** Kaplan–Meier analysis of mammary tumor onset in *MMTV-NIC* mice with wild-type *Ezh2* alleles (*NIC/Ezh2*[+/+]) and mice heterozygous (*NIC/Ezh2*[+/L]) or homozygous (*NIC/Ezh2*[L/L]) for the conditional *Ezh2* allele. Inset table shows group sizes (n), average time to tumor onset and penetrance of mammary tumors. **b** Immunofluorescence (IF) analysis of ErbB2 and Ezh2 expression in mammary glands from 16 week-old *NIC/Ezh2*[+/+] and *NIC/Ezh2*[L/L] mice. **c** IF analysis of ErbB2 and Ki67 expression in mammary glands from 16 week-old *NIC/Ezh2*[+/+] and *NIC/Ezh2*[L/L] mice. **d** Quantification of Ki67-positive nuclei in ErbB2-expressing cells in mice of both genotypes. Data in (**b**–**d**) represent analyses of 4 independent mice per genotype, 5 fields of view per mammary gland. All scale bars represent 50 μm. **e** Hematoxylin-stained, whole-mounted mammary glands from *MMTV-NIC* mice treated for 10 weeks with GSK126 or vehicle control. White arrows indicate transformed mammary epithelial lesions. Scale bar indicates 500 μm. **f** IF analysis of ErbB2 and Ki67 expression in mammary glands from *MMTV-NIC* mice treated for 10 weeks with GSK126 or vehicle control. Scale bar indicates 100 μm. **g** Quantification of Ki67-positive nuclei in ErbB2-expressing cells in mice as in (**f**) (n = 6 per treatment group). In all scatter plots, the center line indicates the mean and error bars are ±SEM, with * = p < 0.05 (unpaired, two-tailed Student's *t*-test)

targeted specifically. Overall, our data are consistent with previous reports suggesting that c-Src stimulates mitochondrial ATP production[36,37]. Although cells evolving without c-Src in vivo displayed a metabolic shift toward glycolysis, they were unable to alleviate energy stress. Since mTORC1 activity is coupled tightly to energy sufficiency[35], the ability of c-Src to bolster mitochondrial ATP generation facilitates mTORC1-dependent translation of PRC2 components and ErbB2-driven tumorigenesis. In accordance with previous findings[39], we observed reduced expression of the mitochondrial transcription factor Tfam and the ATP synthase (complex V) subunit Atp5o, as well as components of ETC complexes I and IV, in c-Src-deficient cells. Thus, inhibition of mTORC1 due to c-Src loss may exacerbate mitochondrial dysfunction, further contributing to energy stress. The

precise mechanisms by which c-Src promotes OXPHOS remain elusive, as putative mitochondrial substrates are largely inconsistent between different studies and c-Src lacks a mitochondrial targeting sequence (MTS), making its means of accessing mitochondrial targets unclear. Notably, the non-receptor tyrosine kinase c-Abl, also lacking an MTS, reaches mitochondrial substrates through interactions with MTS-containing proteins[59] or phosphorylates substrates exposed at the cytosolic surface of the outer membrane[60]. Future studies will aim to identify and characterize analogous mechanisms mediating metabolic control by c-Src in ErbB2+ breast cancer cells.

In cells where PRC2 exerts pro-tumorigenic functions, linking its expression with energy sensing mechanisms may contribute to metabolic checkpoints that facilitate proliferation when ATP

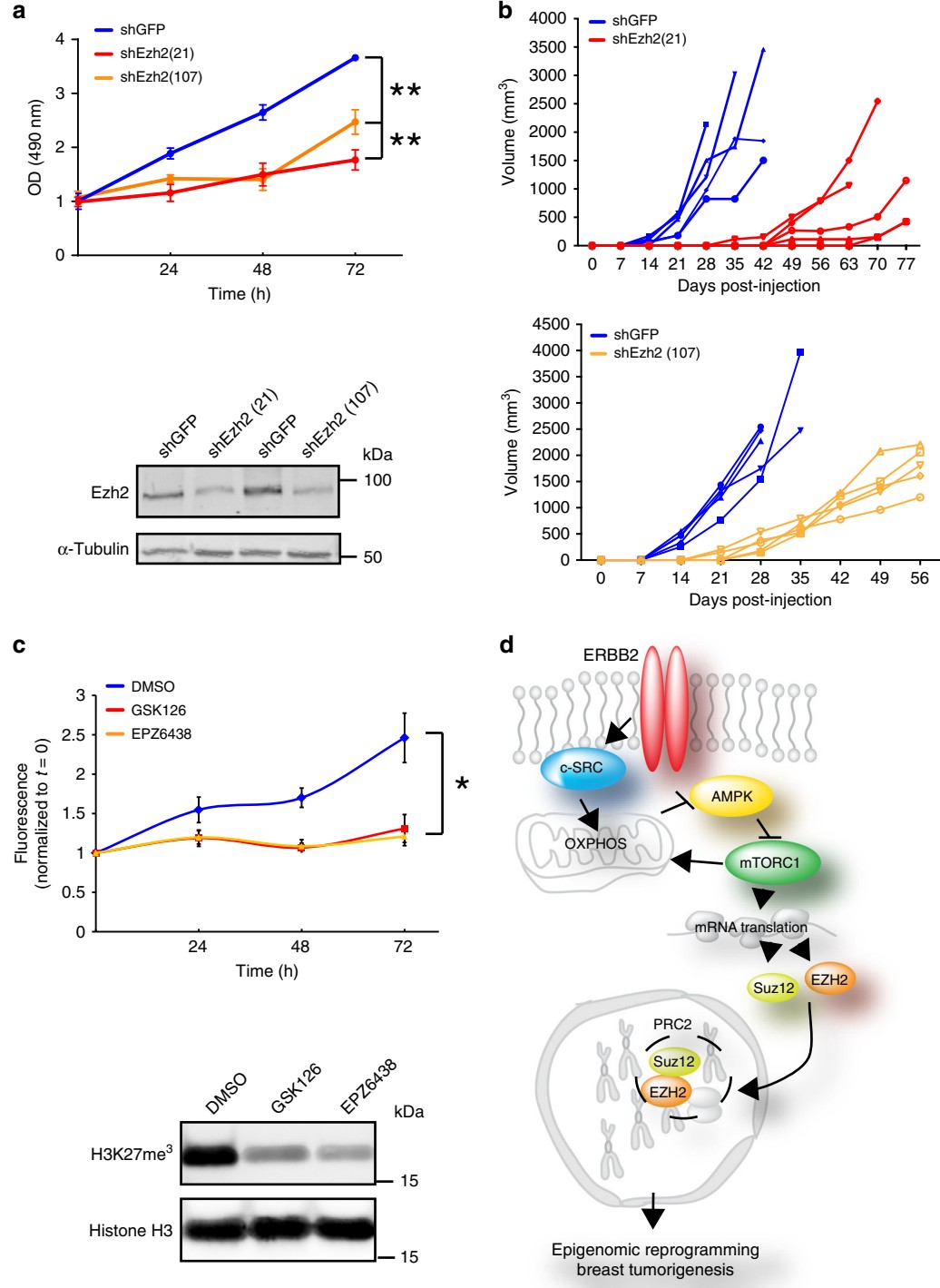

**Fig. 9** Ezh2 silencing or inhibition inhibits ErbB2 + mammary tumor cell growth. **a** Ezh2 expression was silenced in an established ErbB2-driven breast cancer cell line derived from the NIC model using lentiviral shRNA (pLKO.1 shEZH2(21)) and pLKO.1 shEZH2(107). An shRNA targeting GFP was used as a negative control. Top panel shows a representative cell growth assay (mean ± SEM, ** = $p < 0.01$; One-way ANOVA with Dunnett's post-test). Bottom panel is a representative immunoblot showing silencing of Ezh2. **b** Cells as in **a** were injected into the mammary fat pads of athymic nude (NCr) mice. Tumor outgrowth was monitored by palpation and caliper measurement. **c** Upper panel shows proliferation of ErbB2-positive *MMTV-NIC* breast cancer cell lines treated with the EZH2 inhibitors GSK126 and EPZ6438 ($n = 4$ cell lines derived from independent tumors, DMSO vs. GSK126, $p = 0.038$; DMSO vs. EPZ6438, $p = 0.029$; one-way ANOVA with Tukey's post-test). Lower panel is a representative immunoblot showing inhibition of H3K27 tri-methylation. **d** Schematic diagram illustrating c-Src-dependent regulation of PRC2 components via modulation of OXPHOS and mRNA translation

levels are sufficient and suppress proliferation during energy stress. This regulation often involves AMPK, which our results indicate is critical in suppressing mTORC1 activity and PRC2 expression in c-Src-deficient cells, as well as in stimulating

glycolysis, which is another established metabolic output of AMPK activity[61]. We show that OXPHOS is highly responsive to acute targeting of c-Src and its re-expression in deficient cells. Interestingly, however, acute c-Src deletion or kinase inhibition in

wild-type cells did not replicate the up-regulation of glycolysis seen in chronically c-Src-deficient cells, but suppressed baseline levels of glycolysis, which is consistent with previous findings on phosphorylation of glycolytic regulators and enzymes by SFKs[62]. Considered with the evidence for transcriptional up-regulation of glucose metabolism, these data argue that c-Src-deficient tumors acquire enhanced glycolytic capacity through genetic and/or epigenetic changes establishing AMPK-dependent stimulation of glycolysis[61] during tumor evolution in vivo. We also observed significantly reduced steady-state amino acid levels in c-Src-deficient ErbB2+ cells, potentially activating another critical metabolic checkpoint inactivating mTORC1. A recent study has implicated c-Src in amino acid sensing through inhibition of the GATOR1 complex, which inactivates mTORC1 under low amino acid conditions[63]. AMPK activation is also implicated in the response to amino acid starvation via Ca2+ influx and CAMKKβ[46]. Overall, our data indicate that energy stress, rather than amino acid starvation or sensing, is the primary driver of AMPK activation and mTORC1 suppression in c-Src-deficient ErbB2+ cells. However, it is possible that multiple metabolic alterations, including changes in amino acid metabolism, could contribute to these phenotypes.

PRC2 exerts highly context-specific effects in cancer, with activating and loss-of-function mutations of its components supporting oncogenic and tumor suppressive functions, respectively[4]. EZH2 overexpression was linked to the transformed phenotype initially in solid tumors including prostate cancer and breast cancer. However, subsequent studies suggested that Ezh2 is dispensable for breast carcinogenesis[6,64] or that Ezh2 inhibition is tumor-promoting[7]. These inconsistencies may stem from breast cancer heterogeneity, which presents a variety of contexts that determine the requirement for PRC2 activity. For example, Ezh2 appears to be dispensable for tumorigenesis in some models of TNBC[7,64]. Here, in contrast, we show that Ezh2 is required for tumor progression beyond an early hyperplastic stage in models of ErbB2-positive breast cancer and for proliferation and tumor growth in established ErbB2+ breast cancers. Such differences between breast cancer subtypes may be explained by differing cells of origin[4], which, in conjunction with interactions between subtype-specific genetic events, signaling and metabolism, result in distinct requirements for PRC2-mediated gene silencing. Transcriptomic and epigenomic profiling confirmed the significant loss of PRC2 function in c-Src-deficient tumor cells. Interestingly, our analyses revealed that the vast majority of H3K27me3 in NIC/c-Src+/+ cells occurs in non-coding regions of the genome, while few PRC2-silenced protein-coding genes in control cells were transcriptionally activated in c-Src-deficient cells. This may indicate that c-Src-deficient cells have adopted alternative mechanisms to maintain transcriptional silencing of genes originally targeted by PRC2. Regulation of non-coding RNAs, including retrotransposons and repeat regions, is increasingly recognized as an important function of PRC2[65,66] and is an interesting avenue for further study in the context of breast cancer.

While EZH2 is an emerging therapeutic target in cancer[52], it also plays vital roles in normal stem cells and tissue homeostasis[1,4]. Identifying mechanisms underlying EZH2 overexpression in cancer, such as those described here, may allow selective targeting of aberrant PRC2 activity while minimizing toxicity in normal tissues. Importantly, the diversity of PRC2 functions in different cancers necessitates careful characterization of its roles in specific tumor types to identify appropriate contexts for targeting PRC2, either directly or through its upstream regulators. Our findings suggest that such strategies may be beneficial for patients with ERBB2-overexpressing tumors, where lack of response and the development of resistance to ERBB2-targeted therapies present significant obstacles to improving patient outcomes.

## Methods

**Animal models.** All experiments involving mice were carried out under an Animal Use Protocol approved by the McGill University Downtown Campus Facility Animal Care Committee (FACC), a branch of the McGill University Animal Care Committee (UACC). All experiments were conducted in accordance with McGill University and Canadian Council on Animal Care (CCAC) ethical guidelines. The MMTV-NIC model, *Src* and *Ezh2* conditional mice have been described previously[2,17,18]. Cohorts of female *MMTV-NIC* mice carrying wild-type or conditional alleles of *Src* and *Ezh2* on a pure FVB/N genetic background were monitored for mammary tumor formation by twice weekly palpation. Female littermates were group housed under SPF conditions with a 12 h day/night cycle and ad libitum access to food and water. Once detected, tumors were measured weekly using calipers until they had reached a volume of 2.5 cm$^3$ in size for a single mass or a total volume of 5 cm$^3$ for multifocal tumors, at which point mice were euthanized in accordance with approved facility protocols. For studies of Ezh2 targeted therapy in the MMTV-NIC model (Fig. 8), mice were used prior to the onset of palpable tumors (treatment beginning at 10 weeks of age).

For Patient-Derived Xenograft models, the recipients were female *NOD/SCID/gamma* (NSG) immunocompromised mice (purchased from Charles River). For propagation, tumors were excised, cut into pieces of ~8 mm$^3$ and re-implanted into the left inguinal mammary fat pads of *NSG* mice.

For orthotopic transplantation of cell lines derived from the MMTV-NIC tumor model, the recipients were female *NCr* athymic nude immunocompromised mice (purchased from Taconic). In total $5 \times 10^5$ tumor cells were suspended in 30 μl of PBS and injected unilaterally into the left inguinal fat pad. Tumor growth was measured by twice-weekly caliper measurements.

**Human subjects.** For TMA studies, two cohorts of breast cancer patients were used. All patients provided written, informed consent. The studies were approved by the Research Ethics Office/Institutional Review Board (IRB) of the Faculty of Medicine, McGill University (Study #A07-M40-15B) and of the Royal Prince Albert Hospital (HREC/15/RPAH/531 & X15-0388) and the Garvan Institute/St. Vincent's Hospital (HREC/08/CIPHS/62). The Garvan/St.Vincent's Hospital cohort consists of 292 female breast cancer patients with a median age of 54 years who were diagnosed with invasive ductal adenocarcinoma between February 1992 and August 2002. Formalin-fixed, paraffin-embedded (FFPE) tissue blocks were retrieved from St Vincent's Public Hospital (Sydpath) and St Vincent's Private Hospital (Douglas Hanly Moir Pathology), Sydney, Australia, and used to construct TMAs consisting of 2 cores of approximately 1 mm diameter per patient (approximately 80 cores per slide). Detailed information regarding the clinicopathologic characteristics of this cohort has been published elsewhere[67].

The Royal Prince Alfred Hospital (RPAH) HER2-positive cohort was derived from a larger cohort of 700 patients diagnosed with invasive ductal adenocarcinoma at the Royal Prince Albert Hospital, Sydney, Australia[68]. FFPE tumor samples and patient files were retrieved from the Department of Tissue Pathology and Diagnostic Oncology, Royal Prince Alfred Hospital and Concord Repatriation General Hospital tumor archives (Sydney, Australia) and were reviewed by a specialist breast pathologist. A HER2+ subset consisting of 131 female breast cancer patients with a median age of 53.61 years was selected and TMAs were constructed using three 1 mm cores taken from different areas of each tumor. HER2 status was confirmed via immunohistochemistry. IHC = 0 or 1, 2+ cases were confirmed by HER2 fluorescence in-situ hybridization (FISH).

For matched immunofluorescence (IF) and mRNA (RNA-Seq) analysis of patient samples (Fig. 4), frozen sections of primary tumor biopsy material from a cohort of HER2+ patients were obtained and previously published RNA-Seq data were used[22,23]. Patients were from two independent, multi-site clinical trials (03-311, NCT00148668; and BrUOG 211B, NCT00617942) and had provided prior written, informed consent. Tissue collection for this study was approved by the Institutional Review Boards (IRBs) of the Dana Farber Cancer Institute and the Yale Comprehensive Cancer Centre (03-311) or the Brown University Oncology Group, the Yale Comprehensive Cancer Centre and the City of Hope Comprehensive Cancer Center (BrUOG 211B).

PDX models were established from tumor samples collected with informed consent from hospitals of the McGill University Health Centre, Montreal, QC. Tissue collection was performed under a protocol approved by the Research Ethics Office/Institutional Review Board (IRB) of the Faculty of Medicine, McGill University.

**Primary cell cultures and cell lines.** This study used primary cultures and cell lines derived from mammary tumors arising in female *MMTV-NIC* transgenic mice[18]. Tumors at 8 weeks post palpation were processed with a McIlwain tissue chopper (Mickle Laboratory Engineering), dissociated in collagenase B (Roche, 11088831001) and Dispase II (Roche, 4942078001) (2.4 mg/ml each) for 1 h at 37 °C, washed three times with PBS/1mM EDTA and plated in Complete Media, consisting of DMEM (Wisent, 319-005-CL) supplemented with 5% FBS (Wisent, 080-150), 5 ng/ml EGF (Wisent, 511-110-UM), 1 μg/ml Hydrocortisone (Sigma,

H4001), 5 μg/ml Insulin (Wisent, 511-016-UG) and 35 μg/ml Bovine Pituitary Extract (BPE – Hammond CellTech, 1078-NZ). Cells were grown in a humidified, 5% $CO_2$, 37 °C incubator in Complete Media. The genotype of murine cell lines was authenticated by PCR on genomic DNA to detect the presence of the MMTV-NIC transgene and the presence of wild-type and LoxP-flanked conditional alleles (oligonucleotide details in Supplementary Data 1) and also by immunoblotting to detect the expression of ErbB2, Cre recombinase and c-Src proteins. Cells were routinely tested (every 2 weeks) for mycoplasma using the MycoAlert kit (Lonza, LT07-118) according to the manufacturer's instructions. All cells used in this study were negative for mycoplasma contamination.

**Human cell lines**. Lentiviruses were produced using 293T cells purchased from ATCC. These cells were used at early passage (less than 10 passages) and were not authenticated. SkBr3, MDA-MB-361, and MDA-MB-231 (human breast cancer) cells were purchased from ATCC, used at early passage and were not authenticated. Cells were routinely tested (every 2 weeks) for mycoplasma using the MycoAlert kit (Lonza) according to the manufacturer's instructions. All cells used in this study were negative for mycoplasma contamination.

**Lentiviral Transduction**. Lentiviruses bearing shRNA against Ezh2 and Eif4ebp1 were produced in 293T cells (ATCC) co-transfected with the vectors pMD2.G and psPAX2, gifts from Dr. Didier Trono (Addgene plasmids #12259 and #12260). Lipofectamine 3000 (Invitrogen, L3000075) was used for transfection according to the manufacturer's instructions and virus-containing media was harvested and filtered through a 0.45 μm filter at 24 and 48 h post-transfection. MMTV-NIC cells were transduced in the presence of 10 μg/ml polybrene (Sigma, 107689). Transduced cell lines were selected and maintained in Complete Media with 2 μg/ml puromycin (BioShop, PUR333).

**siRNA Transfection**. MMTV-NIC cells were transfected with siRNAs targeting Lkb1 or a negative control siRNA (see Supplementary Data 1 for oligo information) at a final concentration of 5 nM using Lipofectamine 3000 according to the manufacturer's instructions. Cells were lysed for immunoblotting 72 h after transfection.

**Drug Treatment of Cell Lines**. Everolimus (LC Laboratories, E-4040), Torin 1 (LC Laboratories, T-7887), and silvestrol (MedChem Express, HY-13251) were reconstituted in DMSO and cells were treated at the concentrations indicated in the figures for 24 h. Phenformin (MedChem Express, HY-16397A) and eCF506[32] (provided by Dr. Asier Unciti-Broceta) were reconstituted in sterile water, while rotenone (Toronto Research Chemicals, R700580), oligomycin A (Sigma, 75351), CCCP (Tocris, 04-525-00), GSK621 (Adooq, A15896), PF-06409577 (Tocris, 6114), and Dasatinib (LC Laboratories, D-330) were reconstituted in DMSO. Cells were treated at the concentrations indicated in the figures for 24 h. For treatment with GSK126 (Custom synthesis, Mercachem) and EPZ6438 (MedChem Express, HY-13803) (both reconstituted in DMSO), cells were treated at a concentration of 2 μM for 7 days, with media and drug replenished on day 3 and day 6. Cells were seeded into 96-well plates for use in proliferation assays or histones were extracted for immunoblotting analysis on day 7.

**Preclinical studies in PDX models**. Following transplantation of PDX tumors, mice were randomly assigned to treatment groups and monitored for tumor growth by twice-weekly palpation. Once tumors had reached a size of 5 mm × 5 mm (~65 mm³) treatment was initiated. Dasatinib (10 mg/kg) and AZD2014 (20 mg/kg – MedChem Express, HY-15247) were formulated in 30% Sulfobutylether-β-cyclodextrin (MedChem Express, HY-17031) in water and administered by daily oral gavage. Mice were weighed twice weekly and doses were adjusted according to bodyweight. Tumor growth was measured by twice-weekly caliper measurements. Drug administration, data collection and data analysis were performed by separate individuals who were blinded with respect to the treatment group of each mouse.

**Preclinical studies in GEMMs**. 10 week-old MMTV-NIC mice were randomly assigned to receive GSK126 (300 mg/kg, dissolved in 20% sulfobutylether-β-cyclodextrin in water) or vehicle control (20% sulfobutylether-β-cyclodextrin in water) by intraperitoneal injection every 48 h. Mice were euthanized at 20 weeks of age, following 10 weeks of treatment. At this time point, MMTV-NIC mice typically exhibit pre-malignant mammary epithelial lesions and small, pre-palpable mammary tumors. Drug administration, data collection and data analysis were performed by different individuals who were blinded with respect to the treatment group of each mouse.

**Histology, Immunostaining and Tissue Microarrays (TMA)**. Tumors or mammary glands were fixed in 10% neutral buffered formalin (Leica, 3800600) for 24 h and embedded in paraffin for sectioning. Sections were cut at 4um, deparaffinized in xylene and antigen retrieval was performed with 10 mM sodium citrate (pH 6) using a pressure cooker. Sections were then blocked with 10% Power Block (BioGenex, HK083) in PBS for 10 min at room temperature. Sections were incubated with primary antibody at 4 °C overnight. For immunofluorescence, sections

were incubated with secondary antibodies for one hour at room temperature, followed by DAPI (4′,6-Diamidino-2-Phenylindole, Dihydrochloride, Thermo-Fisher, D1306) for 15 min, washed three times in PBS and mounted in Immu-Mount (Thermo Scientific, 9990412). Immunostained samples were imaged using a Zeiss LSM800 confocal microscope and analyzed with ZEN software. For quantification of Ki67 positivity (Fig. 8d), five independent fields of view in mammary gland sections from four mice per genotype were analyzed. A total of 2600 cells in ErbB2-positive lesions from NIC/Ezh2$^{+/+}$ mice and 1200 cells in ErbB2-positive lesions from NIC/Ezh2$^{L/L}$ mice were counted. For quantification of Ezh2 positivity in PDX models (Fig. 4g) and Ki67 positivity in MMTV-NIC mice receiving GSK126 or vehicle control (Fig. 8g), HALO software (Indica Labs) was used according to the manufacturer's instructions to specifically analyze human pan-cytokeratin-positive tumor cells in PDX models (Fig. 4g) or ErbB2-positive mammary epithelial cells in MMTV-NIC mammary glands (Fig. 8g), excluding stroma and areas of necrosis. A minimum of 5000 (PDX models) or 1000 (MMTV-NIC mammary glands) cells per mouse was counted.

For IHC, sections were deparaffinized and blocked as above and endogenous peroxidase activity was quenched by incubation in 3% hydrogen peroxide for 20 min. Sections were incubated with primary antibody overnight at 4 °C, washed three times in PBS and then incubated with ImmPRESS HRP polymer reagents to detect rabbit and mouse antibodies (Vector Elite, MP-7401 and MP-7402, respectively) according to the manufacturer's instructions. After three further washes in PBS, IHC staining was visualized using the SignalStain DAB Substrate kit (Cell Signaling, 8059S) according to the manufacturer's instructions. Sections were then counterstained with hematoxylin, dehydrated, and mounted with Clearmount (Invitrogen, 10058832). Images were acquired using an Aperio-XT slide scanner and analyzed using a nuclear staining algorithm in the associated software (Aperio Technologies).

TUNEL (Apoptag Peroxidase In Situ Apoptosis Detection Kit, EMD Millipore, S7100) was performed on tissue sections as described above and according to the manufacturer's instructions

For IHC analysis of TMA slides, four-micron sections were cut from each TMA, mounted on SuperFrost Plus glass slides and baked for 2 h at 79 °C. IHC was performed as described above. Nuclear (EZH2) and cytoplasmic (phospho-EIF4EBP1 and phospho-Src family kinases) staining was assessed by an experienced breast pathologist (S.A.O.). For each marker, both a staining intensity score (0: negative, 1+: weak, 2+: moderate and 3+: strong) and a percentage of positively stained cells were assigned to each core. For each core, a histoscore (H score) was derived by multiplying the intensity score by the percentage of positively stained cells for each marker. To determine correlations between the histoscores for the markers, a Spearman's rank correlation test was applied.

The following primary antibodies were used for immunostaining techniques at the indicated dilutions: Ki-67 - Abcam, ab38113, 1/500; BrdU - Cell Signaling, 5292, 1/100; H3K27me3 (C36B11) - Cell Signaling, 9733, 1/100; Ezh2 (D2C9) XP - Cell Signaling, 5246, 1/500; ErbB2/c-Neu (AB3) - Calbiochem, OP15, 1/100; P-4E-BP1 Thr37/46 - Cell Signaling, 2855, 1/100; P-Rps6 Ser 240/244 - Cell Signaling, 5364, 1/500; P-AMPK Thr172 - Cell Signaling, 2535, 1/50; P-ACC Ser79 - Cell Signaling, 3661, 1/100; P-Src Family Kinase Tyr416 - Cell Signaling, 2101, 1/100.

The following secondary antibodies were used for immunostaining techniques: For IHC –ImmPRESS HRP polymer reagents as described above, For IF - Alexa Fluor 488 Donkey anti-Rabbit – Fisher Scientific, A21206; Alexa Fluor 555 Donkey anti-Rabbit – Fisher Scientific, A31572; Alexa Fluor 488 Donkey anti-Mouse – Fisher Scientific, A21202; Alexa Fluor 555 Donkey anti-Mouse – Fisher Scientific, A31570; Alexa Fluor 647 Donkey anti-Mouse – Fisher Scientific, A31571; Alexa Fluor 647 Goat anti-Guinea pig – Fisher Scientific, A21450. All fluorescent secondary antibodies were used at a dilution of 1/1000.

**Mammary gland whole mounts**. Mammary glands (number 4, inguinal) were excised, placed on glass slides and fixed overnight in acetone. Glands were stained in haematoxylin for 24 h, destained in 70% ethanol/1% HCl, washed in 100% ethanol, dehydrated overnight in xylenes, and mounted using Permount (Fisher Scientific, SP15-100).

**Protein extraction and immunoblotting**. Freshly excised tumor tissue was immediately flash-frozen in liquid nitrogen, crushed with a mortar and pestle under liquid nitrogen, allowed to thaw briefly and then lysed in ice-cold RIPA buffer (Tris–HCl 50 mM, pH 7.4, sodium chloride 150 mM, 1% Nonidet P-40, 1% sodium deoxycholate, 0.1% SDS, 2 mM EDTA, 0.5 mM AEBSF (Santa Cruz, sc-202041), 25 mM β-glycerophosphate (Sigma, G5422), 1 mM sodium orthovanadate (BioShop, SOV664), and 10 mM sodium fluoride (Sigma, S7920)). Cultured cells were lysed on ice in RIPA buffer. For immunoblottting of histones, the Episeeker Histone Extraction Kit (Abcam, ab113476) was used to extract histones from 5 × 10⁶ cells according to the manufacturer's instructions. Protein concentrations were determined by Bradford assay (Bio-Rad, 5000006) and 40 μg of total protein was analyzed by immunoblot. A Li-COR Odyssey system (Li-COR Biosciences) was used for fluorescent immunoblotting and quantification was performed using associated software. Uncropped images of immunoblots are presented in Supplementary Fig. 11. The following primary antibodies were used for immunoblotting at the indicated dilutions: H3K27me3 (C36B11) - Cell Signaling, 9733, 1/1000; Ezh2 (D2C9) XP - Cell Signaling, 5246, 1/1000; ErbB2/c-Neu (AB3) - Calbiochem,

OP15, 1/500; P-Rps6 Ser 240/244 - Cell Signaling, 5364, 1/1000; P-AMPK Thr172 - Cell Signaling, 2535, 1/500; P-ACC Ser79 - Cell Signaling, 3661, 1/500; P-Src Family Kinase Tyr416 - Cell Signaling, 2101, 1/500; α-Tubulin - Cell Signaling, 2144, 1/2000; β-Actin: Sigma-Aldrich, A5316, 1/2500; c-Src (clone GD11) - Millipore, 05-184, 1/1000; 4E-BP1 - Cell Signaling, 9644, 1/1000; P-4E-BP1 Ser65 - Cell Signaling, 9451, 1/500; Rps6 - Cell Signaling, 2317, 1/1000; GAPDH - Novus, NB300-322, 1/2000; Suz12 - Cell Signaling, 3737, 1/1000; EED - Millipore, 05-1320, 1/500; RbAp46 - Cell Signaling, 6882; RbAp48 - Abcam, ab79416; AMPKα - Cell Signaling, 5832; ACC - Cell Signaling, 3676, 1/1000; Lkb1 - Cell Signaling, 3050, 1/1000; α-Tubulin - Cell Signaling, 2125, 1/2000; Raptor - Cell Signaling, 2280, 1/500; P-Raptor Ser792 - Cell Signaling, 2083, 1/250; Histone H3 - Cell Signaling, 14269, 1/1000; TSC2 - Cell Signaling, 4308, 1/1000; P-TSC2 Thr1462 - Cell Signaling, 3617, 1/500; P-TSC2 Ser1387 - Cell Signaling, 5584, 1/500; P-ErbB2 Tyr1221/1222 - Cell Signaling, 2243, 1/500; P-ErbB2 Tyr877 - Cell Signaling, 2241, 1/500; P-ErbB2 Tyr1248 - Cell Signaling, 2247, 1/500; EGFR - Cell Signaling, 2232, 1/1000; P-EGFR Tyr1068 - Cell Signaling, 3777, 1/500; ErbB3: Cell Signaling, 12708, 1/500; P-ErbB3 Tyr1289 - Cell Signaling, 4791, 1/500; Akt - Cell Signaling, 2920, 1/1000; P-Akt Ser473 - Cell Signaling, 4060, 1/1000; ERK1/2 - Cell Signaling, 9102, 1/1000; P-ERK1/2 Thr202/Tyr204 - Cell Signaling, 9101, 1/1000; Puromycin - Millipore, MABE343, 1/1000; eIF2α - Cell Signaling, 2103S, 1/1000; P-eIF2α Ser51 - Cell Signaling, 3398S, 1/1000; mTOR - Cell Signaling, 2983, 1/1000; P-mTOR Ser 2448 - Cell Signaling, 5536, 1/1000; PRAS40 - Millipore, 05-1070, 1/1000; P-PRAS40 Thr246 - Cell Signaling, 2997, 1/1000; mLST8 - Cell Signaling, 3274, 1/1000. Secondary antibodies used in immunoblotting at a dilution of 1/10 000 were IRDye 800CW Donkey anti-rabbit (LI-COR Biosciences, 925-32213) and IRDye 680CW Donkey anti-mouse (LI-COR Biosciences, 926-68072).

**Immunoprecipitation**. Cells were lysed in ice-cold RIPA buffer and lysated cleared by centrifugation at 4 °C, 15,000 × g for 10 min. Protein concentrations were determined by Bradford assay and Ezh2 was immunoprecipitated overnight at 4 °C with end-over-end mixing from 500 μg of total cellular protein using 5 μl of antibody (Ezh2 (D2C9) XP - Cell Signaling, 5246). Immunoprecipitates were incubated with PureProteome protein A/G magnetic beads (Millipore, LSKMA-GAG10) for 1 h and then washed five times in lysis buffer before analysis by SDS PAGE and immunoblotting. Normal rabbit IgG (Cell Signaling, 2729) was used as a negative control antibody for immunoprecipitations.

**Puromycin incorporation assay**. For puromycin incorporation assays[24], cells were treated with puromycin (10 μg/ml) for 15 min, lysed and processed for immunoblotting or immunoprecipitation as described above, with the exception that, for immunoblotting, the Revert total protein stain (Li-COR Biosciences, 926-11010) was used to quantify the total amount of protein present in each lane of the gel (Li-COR Odyssey system, Li-COR Biosiences) prior to blocking and primary antibody incubation (Millipore, MABE343, 1/1000) to detect puromycin incorporation.

**Quantitative Reverse Transcriptase-Polymerase Chain Reaction**. Total RNA was extracted from flash frozen mammary tumors using an RNeasy Mini Kit (Qiagen, 74106). cDNA was prepared by reverse transcribing the isolated RNA using M-Mulv Reverse Transcriptase, Oligo-dT(23VN) and murine RNase inhibitor (ProtoScript First Strand cDNA Synthesis Kit, New England Biolabs, E6300). Real-time quantitative PCR was performed using LightCycler 480 SYBR Green I MasterMix (Roche, 04887352001) and LightCycler 480 instrument (Roche) and analyzed using associated software.

**Microarray data acquisition and analysis**. Affymetrix GeneChip Mouse Gene 2.0 ST arrays were used to analyze gene expression in c-Src-deficient and control mammary tumors. Total RNA was extracted from 4 *NIC/c-Src*[+/+] and 4 *NIC/c-Src*[L/L] tumors at tumor burden endpoint using the RNeasy Mini Kit and quantified using a NanoDrop Spectrophotometer ND-1000 (NanoDrop Technologies, Inc.) RNA integrity was assessed using a Bioanalyzer 2100 (Aglient Technologies). Sense-strand cDNA was synthesized from 100 ng of total RNA and fragmentation and labeling were performed to produce ssDNA with the GeneChip WT Terminal Labeling Kit according to manufacturer's instructions. After fragmentation and labeling, 3.5 μg DNA was hybridized on GeneChip Mouse Gene 2.0 ST arrays and incubated at 45$^0$C in the Genechip Hybridization oven 640 for 17 h at 60 rpm. GeneChips were then washed in a GeneChips Fluidics Station 450 using Hybridization Wash and Stain kit according to the manufacturer's instructions and scanned on a GeneChip scanner 3000. All procedures were performed at the Genome Quebec Innovation Center, McGill University using arrays, kits and apparatus from Affymetrix. Raw data were first processed to perform gene-level normalization and quality control using Affymetrix Expression Console software. Gene Level Differential Expression Analysis of processed data was performed using Affymetrix Transcriptome Analysis Console software. Differentially expressed genes with ANOVA p-value <0.02 were considered in further analyses. A heatmap of the filtered gene expression dataset was generated using the Hierarchical Clustering module in GenePattern software.

ChIP Enrichment Analysis (ChEA), interrogation of the ENCODE database, KEGG and GO analyses were performed using the EnrichR online tool (http://amp.pharm.mssm.edu/Enrichr/)[69]. Further analysis of gene regulation and pathways

was performed using Ingenuity Pathway Analysis (IPA - Qiagen) according to the manufacturer's instructions.

For analysis of *EZH2* mRNA translational efficiency in MCF7 cells, published microarray data from total cellular mRNA and polysome-associated mRNA[27] were used. These data are available in the GEO database under accession number GSE36847. The effects of metformin, PP242 and rapamycin on mRNA translation were determined using analysis of partial variance (APV) and applied RVM with treatment contrasts, as implemented in the anota R package[70].

**RNA-Seq**. RNA was isolated from two independent *NIC/c-Src*[+/+] and two *NIC/c-Src*[L/L] cell lines, each in biological duplicate, using an RNEasy mini kit (Qiagen, 74106) according to the manufacturer's instructions. RNA concentration was measured using a Qubit® 2.0 Fluorometer (Life Technologies, CA, USA) according to the manufacturer's instructions. Degradation and contamination were initially assessed on 1% agarose gels and purity was determined using the NanoPhotometer® spectrophotometer (IMPLEN). RNA integrity was then assessed using the Bioanalyzer 2100 system (Agilent). In total 3 μg of RNA per sample was used to generate sequencing libraries using the NEBNext® Ultra™ RNA Library Prep Kit for Illumina® (New England Biolabs), following the manufacturer's recommendations, and index codes were added to attribute sequences to each sample. Briefly, mRNA was purified from total RNA using poly-T oligo-attached magnetic beads. Fragmentation was carried out using divalent cations under elevated temperature in NEBNext First Strand Synthesis Reaction Buffer. First strand cDNA was synthesized using random hexamer primers and M-MuLV Reverse Transcriptase (RNase H-). Second strand cDNA synthesis was subsequently performed using DNA Polymerase I and RNase H. Remaining overhangs were converted into blunt ends via exonuclease/polymerase activities. After adenylation of 3′ ends, NEBNext Adapters with hairpin loop structure were ligated to prepare for hybridization. In order to select cDNA fragments of preferentially 150–200 bp in length, the library fragments were purified with AMPure XP system (Beckman Coulter). Then 3 μl USER Enzyme (NEB) was used with size-selected, adapter-ligated cDNA at 37 °C for 15 min followed by 5 min at 95 °C before PCR was performed with Phusion High-Fidelity DNA polymerase, Universal PCR primers and Index (X) Primer. PCR products were purified (AMPure XP system) and library quality was assessed on the Agilent Bioanalyzer 2100 system. Clustering of the index-coded samples was performed on a cBot Cluster Generation System using the HiSeq PE Cluster Kit cBot-HS (Illumina) according to the manufacturer's instructions. The library preparations were then sequenced on an Illumina Hiseq platform and 125 bp/150 bp paired-end reads were generated. Raw data (raw reads) in fastq format were processed through in-house perl scripts to remove reads containing adapter or poly-N and low quality reads from raw data. At the same time, Q20, Q30, and GC content the clean data were calculated. Cleaned data were mapped to the genome of the mouse strain FVB/N. An index of the reference genome was built using Bowtie v2.2.3 and paired-end clean reads were aligned to the reference genome using TopHat v2.0.12. HTSeq v0.6.1 was used to count the reads numbers mapped to each gene, with FPKM of each gene calculated based gene length and read counts (FPKM - expected number of Fragments Per Kilobase of transcript sequence per Million base pairs sequenced) were calculated to estimate gene expression levels. Differential expression analysis was performed using the DESeqR package (1.18.0) and the resulting *P*-values were adjusted using Benjamini and Hochberg's approach for controlling the false discovery rate. Genes with an adjusted *P*-value < 0.05 found by DESeq were assigned as differentially expressed. Analysis of transcriptional regulation and pathway representation in differentially expressed genes from RNA-Seq was performed using EnrichR and IPA as described above.

**ChIP-Seq**. ChIP-Seq to identify and quantify H3K27me$^3$-bound genomic regions was performed using a *Drosophila melanogaster* chromatin spike-in strategy[71]. In total 5 μg of anti-H3K27me$^3$ antibody (H3K27me3 (C36B11) - Cell Signaling, 9733) was immobilized overnight at 4 °C on 20 μl of Magna ChIP Protein A+G magnetic beads (Millipore, 16-663) diluted in 250 μL of PBS + 0.5% BSA and then washed 3 times with PBS + 0.5% BSA. *MMTV-NIC* cells in 15 cm plates were fixed with a 1% final concentration of formaldehyde for 5 min at room temperature and then lysed and sonicated. Equal amounts of chromatin from 2 independent cell lines of each genotype were diluted in 2.5X ChIP dilution buffer (EDTA 2 mM, NaCl 100 mM, Tris 20 mM, Triton 0.5%) + 100 μL of PBS + 0.5% BSA and 750 ng of *D. melanogaster* chromatin (Active Motif, 53083) and 0.4 μg of a *D. melanogaster* H2Av-specific antibody (Active Motif, 39715) were added to each sample. Samples were then added to the antibody-bound beads and left to rotate overnight at 4 °C. Next, beads were washed 3 times for 3 min at 4 °C with 1 mL LiCl buffer (Tris 100 mM, LiCl 500 mM, Na-deoxycholate 1%) then once with 1 mL TE buffer. DNA was eluted with 150 μL of elution buffer (0.1M NaHCO3, 0.1% SDS) overnight at 65 °C. Chromatin immunoprecipitated DNA was purified using a QIAquick PCR purification kit (Qiagen, 28106) and eluted in 35 μL of elution buffer. Purified DNA was sequenced using AAA base pair single end sequencing and reported as.fastq files. Data was assayed for quality using FASTQC and processed using trimmomatic. *D. melanogaster* and mouse sequences were aligned to the BDGP6 and mm10 reference genomes, respectively, using BWA. *D. melanogaster* sequence data were then used to normalize mouse sequence data to remove background noise from random reads[71]. PCR bias and other artifacts were removed by using SAMtools and PICARDtools with default parameters and broad peaks were called

through the use of MACS2 and HOMER with default parameters. Data were visualized by creating tag density maps using the Galaxy ComputeMatrix pipeline.

**Cell proliferation assays**. CellTiter Aqueous MTS (Promega, G3580) (Fig. 9a) and CyQuant (Invitrogen, C7026) (Supplementary Figs. 1e and 10d) proliferation assays were performed in accordance with the manufacturer's instructions. For each time point, 5000 cells per well were seeded in quadruplicate in 96-well optical-bottom plates (Nunc).

**Polysome fractionation and analysis**. *NIC* cells were cultured in 15 cm dishes, washed twice with cold PBS containing 100 μg/ml cycloheximide (Sigma, C7698), then scraped into PBS with 100 μg/ml cycloheximide, collected, and lysed in 450 μl of hypotonic buffer (5 mM Tris–HCl [pH 7.5], 2.5 mM $MgCl_2$, 1.5 mM KCl, 100 μg/ml cycloheximide, 2 mM DTT, 0.5% Triton X-100, and 0.5% sodium deoxycholate). Lysates were loaded onto 10–50% (wt/vol) sucrose density gradients (20 mM HEPES-KOH [pH 7.6], 100 mM KCl, and 5 mM $MgCl_2$) and centrifuged at 36,000 rpm (SW 40 Ti rotor, Beckman Coulter, Inc.) for 2 h at 4 °C. Fractionation of gradients with continuous recording of optical density at 254 nm was performed using an ISCO fractionator (Teledyne ISCO). RNA was isolated from each fraction and from total lysate (input) using TRIzol (Invitrogen, 15596018) according to the manufacturer's instructions, processed using an RNEasy MinElute cleanup kit (Qiagen, 74204) according to the manufacturer's instructions, and then analyzed by QRT-PCR as described above.

**Liquid Chromatography/Mass Spectrometry (LC/MS) Analysis**. Cells were washed three times in ice-cold 150 mM ammonium formate and metabolites were extracted in 50% acetonitrile, 25% methanol, 25% water[40]. Targeted metabolite analysis was performed on an Agilent 6430 triple quadrupole mass spectrometer equipped with a 1290 Infinity UPLC system (Agilent). Sample temperature was maintained at 4 °C while solvents and column temperatures were maintained at 10 °C. Metabolites were separated using a 4.0 μm, 2.1 × 100.0 mm Cogent Diamond Hydride column (MicroSolv Technology) operating at a flow rate of 0.4 ml/min and 5 μl sample injections. Separation solvent A consisted of 15 mM ammonium formate in $H_2O$, pH 5.8 and solvent B consisted of 15 mM ammonium formate in 85% acetonitrile in 15% $H_2O$, pH 5.8. Chromatography started with a 2 min hold at 97% B, followed by a 5 min gradient to 70% B, then washed for 3 min with 98% A, and re-equilibrated to starting conditions for 6 min. Separated metabolites were introduced to the mass spectrometer via electrospray ionization (ESI) operating in either positive or negative ionization mode and were analyzed by previously optimized multiple reaction monitoring (MRM). Quantification was accomplished by comparing MRM peak areas to those from standard calibration curves using MassHunter Quantitive Analysis software (Agilent). MRM transitions in negative ionization mode for quantifying and qualifying ions were 506.0 → 158.9 and 506.0 → 78.9 for ATP, 426.0 → 134 and 426.0 → 79 for ADP, 346.0 → 97 and 346.0 → 78.9 for AMP. Gas temperature and flow were set at 350 °C and 10 L/min. Nebulizer pressure was set at 50 psi and capillary voltage was + 4000 V.

**Gas Chromatography/Mass Spectrometry (GC/MS) Analysis**. Cells were washed three times in ice-cold normal saline solution and water-soluble metabolites were extracted in 80% methanol[40]. Samples dried by vacuum centrifugation overnight at 4 °C were resuspended in 30 μl methoxyamine hydrochloride (MOX) in anhydrous pyridine and added to GC–MS autoinjector vials containing 70μl N-(*tert*-butyldimethylsilyl)-N-methyltrifluoroacetamide (MTBSTFA) derivatization reagent. The samples were incubated at 70 °C for 1 h, after which aliquots of 1 μl were injected for analysis. GC-MS data were collected on an Agilent 5975 C series GC/MSD system (Agilent Technologies) operating in election ionization mode (70 eV) for selected ion monitoring (SIM). The relative amount of each metabolite was determined from the integral ratios of the metabolites to the internal standard and normalized to the number of cells extracted using Agilent MassHunter software. Mass isotopomer distribution was determined using a published custom algorithm[72].

**Respirometry and extracellular acidifcation measurements**. Oxygen consumption and extracellular acidification rates were measured using an XFe96 Extracellular Flux Analyzer (Seahorse Bioscience) using the manufacturer's established protocols. In brief, cells were plated overnight in Seahorse 96-well plates at $1 × 10^4$ per well in 80 μL of complete media. Cells were washed three times in non-buffered DMEM containing 25 mM glucose and 2 mM glutamine and incubated in this medium in a $CO_2$-free incubator at 37 °C for 2 h to allow for temperature and pH equilibration before loading into the XFe96 apparatus. In addition to basal measurements, the Mito Stress Test assay (Seahorse Bioscience, 103015-100) was performed according to manufacturer's instructions. XFe assays consisted of sequential mix (3 min), pause (3 min), and measurement (5 min) cycles, allowing for determination of OCR/ECAR every 10 min. At the end of the assay media was removed from the plate which was then frozen at −80 °C for 24 h. CyQuant dye (Invitrogen, C7026) was then used according to the manufacturer's instructions to generate DNA content values for data normalization. Data were analyzed using Wave software (Seahorse Bioscience).

**Analysis of metabolite levels in conditioned media**. Glucose and lactate levels in conditioned media were measured using a Flux Bioanalyzer (NOVA Biomedical) according to the manufacturer's instructions. For each condition tested, 2 ml of conditioned culture medium from $1 × 10^6$ cells cultured in 6-well plates (Nunc) for 24 h was used. Detached cells and debris were removed by centrifuging media at $15,000 × g$ for 10 min, 4 °C prior to use.

**Quantification and statistical analysis**. Information on group sizes and statistical tests performed for each experiment are in the figure legends. In general, unpaired two-tailed Student's *t*-tests were performed in GraphPad Prism or Microsoft Excel software unless otherwise specified. One-way ANOVA with Tukey's or Dunnett's post-hoc test for multiple comparisons and Kaplan-Meier analysis with logrank tests (Mantel–Haenszel) were performed using GraphPad Prism. Spearman's rank correlation test for analysis of TMA data was performed using Statview (Abacus Systems). Throughout the study, $p < 0.05$ was defined as the threshold for significance.

## Data availability

Gene expression and ChIP-Seq data generated in this study have been deposited in the Gene Expression Omnibus (GEO) database as a SuperSeries with accession number GSE130739 [https://www.ncbi.nlm.nih.gov/geo/query/acc.cgi?acc=GSE130739]. Individual sub-series have the accession numbers GSE93892 (microarray data) [https://www.ncbi.nlm.nih.gov/geo/query/acc.cgi?acc=GSE93892], GSE130661 (RNA-Seq data) [https://www.ncbi.nlm.nih.gov/geo/query/acc.cgi?acc=GSE130661] and GSE130738 (ChIP-Seq data) [https://www.ncbi.nlm.nih.gov/geo/query/acc.cgi?acc=GSE130738]. Published microarray data from total cellular mRNA and polysome-associated mRNA[27] are available in the GEO database under accession number [https://www.ncbi.nlm.nih.gov/geo/query/acc.cgi?acc=GSE36847]. All other data from this study are available within the Article and Supplementary Information or from the corresponding author (WJM) upon request.

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

## Acknowledgements

We thank members of the Muller lab for comments and assistance, Dr. Daina Avizonis and colleagues in the Goodman Cancer Research Centre Metabolomics Core Facility for assistance with metabolomics studies, staff at the Comparative Medicine and Animal Resources Centre (CMARC) for assistance with studies involving mice, Dr. Russell Jones and Dr. Guy Sauvageau for retroviral shRNA constructs targeting AMPK alpha subunits and Ezh2, respectively. We acknowledge funding from the Terry Fox Research Institute (TFRI) Program Project Grant #1048 and the Canadian Institutes of Health Research (to W.J.M., I.T., and V.G.), the US Department of Defense Congressionally Directed Medical Research Programs, Breast Cancer Research Program (W81XWH-11-1-0046) (to H.W.S), the McGill Integrated Cancer Research Training Program (to H.W.S. and A.H.), the Systems Biology Program of the Canadian Institutes of Health Research (to A.H.), FRQ-S Junior 2 award (I.T.), the Réseau de Recherche en Cancer of the FRQS (FRQ-34787) and SU2C (SU2C-AACR-DT-1815) (to M.P.), the Swedish Research Council (to O.L.), the National Breast Cancer Foundation Practitioner Fellowship (PRAC-16-006) and the Sydney Breast Cancer Foundation (both to S.A.O.), CIHR Foundation Award FDN-148373 and a Canada Research Chair in Molecular Oncology (both to W.J.M.).

## Author contributions

Conceptualization: H.W.S. and W.J.M.; Investigation: H.W.S., A.H., V.S.-G., I.N., K.T., M.L., D.Z., C.L., V.P.; Formal Analysis: H.W.S., A.H., C.R.D., S.S., J.R., M.S., E.A., V. v. H., O.L., V.V., S.A.O.; Resources: M.B., P.S., M.P, C.L.C., C.T., N.O.C., A.U.B., A.C.V.C., J.B., E.M., C.S., V.V., L.R.H., S.A.O.; Writing – Original Draft: H.W.S., I.T. and W.J.M.; Writing – Review and Editing: H.W.S., A.H., C.R.D., V.G., I.T., S.A.O., O.L. and W.J.M.; Visualization: H.W.S., A.H., C.R.D., K.T., J.R., V. v. H., I.T.; Funding Acquisition: H.W.S., M.P., V.G., I.T., S.A.O., O.L. and W.J.M.; Supervision: W.J.M.

## Additional information

**Competing interests:** The authors declare no competing interests.

Harvey W. Smith[1,2], Alison Hirukawa[1,2], Virginie Sanguin-Gendreau[1,2], Ipshita Nandi[1,2], Catherine R. Dufour[1,2], Dongmei Zuo[1,2], Kristofferson Tandoc[3], Matthew Leibovitch[3], Salendra Singh[4], Jonathan P. Rennhack[5], Matthew Swiatnicki[5], Cynthia Lavoie[1,2], Vasilios Papavasiliou[1,2], Carolin Temps[6], Neil O. Carragher[6], Asier Unciti-Broceta[6], Paul Savage[1,7], Mark Basik[4,7,8], Vincent van Hoef[9], Ola Larsson[9], Caroline L. Cooper[10,11], Ana Cristina Vargas Calderon[12], Jane Beith[13,14], Ewan Millar[15,16,17], Christina Selinger[18], Vincent Giguère[1,2,7,19], Morag Park[1,2,7,19], Lyndsay N. Harris[4,20], Vinay Varadan[4], Eran R. Andrechek[5], Sandra A. O'Toole[14,21], Ivan Topisirovic[3,19] & William J. Muller[1,2]

[1]Rosalind and Morris Goodman Cancer Research Centre, McGill University, Montréal, QC H3A 1A3, Canada. [2]Department of Biochemistry, McGill University, Montréal, QC H3A 1A3, Canada. [3]Lady Davis Institute for Medical Research, McGill University, Montréal, QC H3T 1E2, Canada. [4]Case Comprehensive Cancer Center, Case Western University, Cleveland, OH 44145, USA. [5]Department of Physiology, Michigan State University, East Lansing, MI 48824, USA. [6]Cancer Research UK Edinburgh Centre, MRC Institute of Genetics and Molecular Medicine, University of Edinburgh, Edinburgh EH4 2XR, UK. [7]Department of Medicine, McGill University, Montréal, QC H3A 1A3, Canada. [8]Department of Surgery, McGill University, Montréal, QC H3A 1A3, Canada. [9]Department of Oncology-Pathology, Science for Life Laboratory, Karolinska Institute, Stockholm 171 76, Sweden. [10]Department of Anatomical Pathology, Pathology Queensland, Princess Alexandra Hospital, Woolloongabba, QLD 4102, Australia. [11]PA Southside Clinical School, School of Medicine, University of Queensland, Brisbane, QLD 4102, Australia. [12]Douglass Hanly Moir Pathology, Macquarie Park, NSW 2113, Australia. [13]Chris O'Brien Lifehouse, Camperdown, NSW 2050, Australia. [14]Sydney Medical School, University of Sydney, Sydney, NSW 2006, Australia. [15]Department of Anatomical Pathology, South Eastern Area Laboratory Service, St George Public Hospital, Kogarah, NSW 2217, Australia. [16]School of Medicine and Health Sciences, University of Western Sydney, Campbelltown, NSW 2560, Australia. [17]Faculty of Medicine, University of New South Wales, Kensington, NSW 2052, Australia. [18]Dept of Tissue Pathology and Diagnostic Oncology, Royal Prince Albert Hospital, Camperdown, NSW 2050, Australia. [19]Department of Oncology, McGill University, Montréal, QC H3A 1A3, Canada. [20]Division of Cancer Treatment and Diagnosis, National Cancer Institute, NIH, Rockville, MD 20890, USA. [21]The Kinghorn Cancer Centre and Cancer Research Program, Garvan Institute of Medical Research, Darlinghurst, NSW 2010, Australia

