## [Peer Review File · Nature Communications]

Reviewers' comments:

Reviewer #1 (Remarks to the Author):

In the manuscript, entitled "An ErbB2/c-Src axis drives mammary tumorigenesis via metabolic reprogramming and translational up-regulation of Polycomb repressor complex2", Smith HW et al. demonstrated a novel role for c-Src on mammary tumorigenesis in the context of hyperactivation of ErbB2. The authors used several approaches (GEMM, PDX and in vitro models) to understand the role of both c-Src and PRC2 in breast cancer. Interestingly, the authors provided compelling evidence for the role of c-Src maintaining the activity of mTORC1, which in turn, promote global translation and upregulation of the Polycomb Repressor complex 2 (PRC2). Furthermore, the authors proved the crucial role of PRC2 in tumorigenesis on ErbB2-driven breast cancer model. Although the authors provided a strong in vivo data that highlights the relevance of c-Src/mTORC1/PRC2 axis in mammary tumors, the mechanism linking c-Src and mTORC1 could be strengthened by performing some additional experiments.

Based on this, I suggest considering the manuscript for publication after minor revision.

General comment:

In general, the manuscript is well presented and is easy to follow the logic behind the experiments. However, I suggest the authors include some additional references in the discussion to compare their results with data previously published.

I suggest to include these articles in the discussion:

Zhang J et al. *Nat Commun.* 2017 Jan 5;8:13732. (c-Src and hexokinase activation)

Poulain L et al. *Leukemia.* 2017 Nov;31(11):2326-2335. (mTORC1 and glycolysis)

Pusapati RV et al. *Cancer Cell.* 2016 Apr 11;29(4):548-562. (mTORC1 and glycolysis)

Eliceiri BP et al. *Mol Cell.* 1999 Dec;4(6):915-24. (C-Src and angiogenesis)

Donnini S et al. *Int J Cancer.* 2007 Mar 1;120(5):995-1004. (c-Src and angiogenesis)

In addition, I suggest correcting the format for some of the references.

Specific comments:

1. Line 113 please amend "tumourwe"

2. Regarding Supplementary Figure 1d, There is some evidence showing that the c-Src play a role in angiogenesis (see articles to be included in the discussion). Considering this, the authors could show an endothelial cells marker in c-Src-proficient and c-Src-deficient tumors. This is important to understand to which extent the lack of c-Src is affecting the environment and how the latter is contributing to tumor growth in this model. This point could be also included in the discussion (line 355).

3. Line 169 and Supplementary Figure 3b. The impairment on the mTORC1 pathway to sense amino acid in the c-Src-deficient cells may be related to a decrease in the amino acids uptake. As a consequence, lower intracellular levels of amino acids may explain the lack of activation of mTORC1 after the addition of amino acids. Does c-Src deficiency affect the expression of amino acid transporters?. The authors should measure the intracellular levels of amino acids in the c-Src-proficient and c-Src-deficient tumors/cells to elucidate whether the inhibition of mTORC1 is related or not to a decrease on the intracellular levels of amino acids.

4 Line 193 please amend "that that".

5. Line 202 please amend "tumouralso".

6. Line 211 please amend "tumourFinally"

7. 222 and 241 The authors showed that the phosphorylation of TSC2 (pThr1462) was not affected, something that was expected given that they did not see differences in phospho-AKT (Ser473). However, authors should check other phosphorylation site on TSC2 (pSer1387) and Raptor (P-Ser792), which are both mediated by the AMPK and impinge on mTORC1 signaling. In addition, amino acids starvation has been described as an activator of AMPK via CaMKKbeta (Ca²⁺/calmodulin-dependent protein kinase kinase-beta, Ghislat G et al. *J Biol Chem.* 2012;287:38625-38636.). Is this contributing to the activation of AMPK in addition to the increased levels of AMP?. These experiments will contribute to clarify and provide a stronger mechanistic link between AMPK and mTORC1 for this model.

8. Line 234-237 the authors claim that the deficiency in c-Src leads to a metabolic rewiring. Is this

a metabolic rewiring induced by the lack of c-Src or the global decrease in the translation decrease also the expression of electron transport chain proteins, which could increase glycolysis?. Authors should measure the levels of the proteins of the electron transport chain to understand if this is connected or not to the decrease in respiration. Is the inhibition of mTORC1 sufficient to decrease the expression of these proteins in this context?. These experiments will provide new insights regarding the decrease in the respiration of c-Src deficient cells.

Reviewer #2 (Remarks to the Author):

Polycomb Repressor Complex 2 (PRC2) has been shown to act either as an oncogene or as a tumour suppressor depending on the cellular context and tumour type. Components of the PRC2 complex (SUZ12, EZH2) have been shown to be highly expressed in certain cancer types, particularly in ERBB2-positive breast cancers (which also have global enrichment of H3K27me3). However, no mechanisms accounting for upregulation of PRC2 components (EZH2 and SUZ12) have been reported so far, and there is no evidence for an effect of PRC2 in driving mammary tumorigenesis raising the possibility that the high levels of H3K27me3, EZH2, and SUZ12 might be only correlative and not causal. This paper aims to address these outstanding questions using pathophysiologically relevant approaches. This is a real tour de force.

By demonstrating a complex and original interplay between c-Src oncogenic signaling, metabolic reprogramming, and epigenetic regulation, the authors propose a mechanism for how PRC2 components (EZH2 and SUZ12) are stabilised through regulation of the mRNA translation, and how PRC2 participates in ERBB2-driven tumorigenesis. Mechanistically, the authors showed that c-Src ablation impairs ERBB2-driven mammary tumorigenesis, and that c-Src promotes mTORC1 activation. Ultimately, this results in an increase in global mRNA translation and an increase in Ezh2 and Suz12 protein levels. Importantly c-Src ablation does not result in changes in the phosphorylation status of key signaling molecules such as Akt, Erk, Egfr, Erbb2, and Erbb3, suggesting a non-canonical mechanism of mTORC1 activation. c-Src ablation was shown to cause a reduction in oxidative phosphorylation and ATP reduction, leading to the activation of AMPK (a potent inhibitor of mTORC1). Finally, after deciphering mechanism through which Ezh2 is upregulated in Erbb2-driven breast cancer, the authors show that ablation of Ezh2 strongly inhibits tumour growth.

The experiments are well controlled and the article is well-written, with a very detailed material and methods section. Many aspects of the research are thorough and well-discussed e.g. a good characterisation of mammary gland development of Ezh2 deletion in the mouse strain used. However, the overall narrative of the paper is sometimes difficult to follow, as it attempts to cover a lot of complex topics, and the connections between these topics are not immediately apparent. A small schematic summary of the results, incorporated into the final figure of the paper, would assist with this understanding. The authors should address the questions below for the manuscript to become suitable for publication.

Major points

1. Figure 3 addresses the reduction in mTORC1 activity that occurs with Src deletion. The authors claim that c-Src enhances mTORC1 activity, which in turn increases the synthesis of proteins including Ezh2 and Suz12. To support their claim, they show immunoblotting of downstream targets of mTORC1 phospho-Rps6 and phospho-Eif4ebp1; however, there is no western blot directly showing the total/phospho levels of mTORC1 components. The authors should perform a western blot showing total and phosphorylation status of mTORC1 components (e.g. mTOR pSer2448) in control NIC/c-src+/+ versus NIC/c-src-/- to address this question. mTORC1 component blots should be added to figure 3a as well as figures 4d and 5h (to address whether AMPK activation results in mTORC1 inhibition and not only in inhibition of downstream targets of

mTORC1).

2. DNA microarrays and analysis (Figure 1d): the focus is purely on the up-regulated genes in the Src-deleted tumours. There is no discussion of the down-regulated genes, even though there are in fact more down-regulated than upregulated genes. There is a lack of functional analysis of the up/down-regulated genes (with the exception of glucose metabolism enrichment plot in figure 5g) – authors state that c-Src loss significantly impedes Ebb2-driven mammary tumorigenesis but there is no exploration of the pathways/functions affected by c-Src loss. Are there more gene sets that are enriched that could explain the metabolic effects of c-Src deletion?

Similarly, for the H3K27me3 ChIP-seq (figure 1f) there is no functional analysis of the peaks that are regulated by c-Src expression: the peaks are nicely centred around the TSS and could be easily correlated with changes in gene expression. This would provide another piece of evidence that the changes in gene expression are actually being caused by H3K27me3 loss due to a downregulation of PRC2 activity, and a functional readout of how c-Src alters cellular behaviour to promote tumorigenesis in Ebb2-driven tumours. At the moment the evidence provided for the gene expression changes being caused by PRC2 (downstream of c-Src deletion) is purely computational (ChEA and Encode analyses). Are there genes that change in expression that are not correlated with H3K27me3 loss that could explain some of the driving metabolic effects of c-Src deletion? Does re-expression of PRC2 sub-units such as Ezh2 and Suz12 rescue a subset of the gene expression changes?

3. In figure 1, the authors show that Src deletion impairs ErbB2-driven tumorigenesis; similarly, in the last figure they show that ablation of Ezh2 decreases ErbB2-driven tumorigenesis. In the title of the paper the authors claim that ErbB2/c-Src axis drives mammary tumorigenesis via metabolic reprogramming and translational up-regulation of PRC2, suggesting a synergistic contribution of c-Src and PRC2 to tumour development. Does treatment of Ebb2-driven tumours with dual-targeting therapy (e.g. dasatinib + Ezh2 inhibitor) have an added benefit over monotherapy (where resistance might be expected to evolve)? Can the authors perform this experiment using the ErbB2-driven breast cancer line derived from NIC model (same setting as figure 8b) ?

4. Not fully addressed is how the metabolic effects of c-Src deletion, which appear to be the kick-start for the whole process of PRC2 inactivation, arise. e.g. is it due to changes in expression of metabolic genes (as demonstrated for glucose metabolism)? What transcription factors downstream of c-Src could be driving these gene expression changes? It is stated in the discussion that the precise mechanisms by which c-Src alters mitochondrial function have not been fully elucidated by other studies, could this point be further elaborated on?

Additional points

5. Add legend to figure 1d to make it clear which genes are up/down-regulated.

6. Relatively small number of H3K27me3 peaks even in the control setting (figure 1f) – how does this compare to other H3K27me3 ChIPs?

7. In relation to figure 2 – the authors state there is “globally reduced translation in Src-deficient cells”

a. However, there is no alteration in the polysome distribution for Eed and Actb; this suggests an element of selectivity for the effects on translation, rather than a global reduction, which is not addressed

b. Why/how is the translation of some transcripts and not others affected?

c. E.g. do they bear a 5'TOP sequence (cytosine at the penultimate nucleotide position followed by a stretch of 4–14 pyrimidines) rendering them particularly sensitive to mTORC1 regulation ? This could also be addressed in the discussion

8. Figure 3c – state in the figure or legend which ERbB2 cell line was used for this experiment

9. The lack of phosphorylation changes for key signaling mediators (e.g. RTKs, Erk, Akt) with Src deletion would be better in a main figure, and if possible introduced earlier to make it clear that this canonical mechanism is not what is driving mTORC1 inactivation with Src deletion

10. Figure 4d – can the authors add an immunoblot of H3K27me3 using the same extract to see whether c-Src and mTORC1 inhibition also result in a reduction in H3K27me3 in the PDX setting?
11. Figure 5d – can the authors specify in the text that pT172 is a sign of AMPK activation?
12. Need to explain further page 11, line 247 - “the biguanide phenformin, which bypasses the input of c-Src”
13. In relation to figure 6:
 - a. Total AMPK and phosphor-AMPK levels should be shown in panel a
 - b. In panel c, why do the c-Src L/L-LacZ cells have lower basal respiration than their +/- counterparts?
14. In the text, line 317, the authors want to address whether EZH2 ablation is required for tumor initiation however they do not provide data to answer this question
 - a. Can the author perform a tumorsphere assay (in Ultra low attachment plate or in 3D matrigel) to see whether EZH2 ablation impacts tumor initiating capacities of HER2 driven breast cancer cell derived from the NIC model?
 - b. Indeed, in the last figure, the delay in tumor growth is more likely due to a decreased proliferation (low Ki67 index) rather than inhibition in tumor initiation

Minor issues noted in abstract and introduction:

- o Line 41 : « However the underlying mechanisms are obscure. » Mechanisms of what ? reformulate
 - o Line 45 : « c-src coordinates bioenergetics with PRC2 overexpression » what does this mean ? Description is vague and does not assist with understanding this complex pathway
 - o Line 55 : lack of reference highlighting a role of PRC2 in pluripotency
 - o Line 68/69 : among the most prominent tumor type..... is breast cancer : reformulate
 - o Line 84 : lack of reference linking epigenetic dysregulation and metabolic reprogramming
 - o Line 87 : upon directly by : reformulate
 - o Line 92/93/94 : not easily understood (provide an example of key signalling intermediates?)
 - o There is a heavy reliance on review articles rather than original research articles to support claims regarding the consequences of metabolic reprogramming on tumorigenesis e.g. lines 83-96
- Spelling/grammatical errors:
 - o Line 113 – remove “tumour”
 - o Line 121 – remove open bracket
 - o Line 202 – remove “tumour”
 - o Line 211 – remove “tumour”

Reviewer 1

In the manuscript, entitle “An ErbB2/c-Src axis drives mammary tumorigenesis via metabolic reprogramming and translational up-regulation of Polycomb repressor complex2”, Smith HW et al. demonstrated a novel role for c-Src on mammary tumorigenesis in the context of hyperactivation of ErbB2. The authors used several approaches (GEMM, PDX and in vitro models) to understand the role of both c-Src and PRC2 in breast cancer. Interestingly, the authors provided compelling evidence for the role of c-Src maintaining the activity of mTORC1, which in turn, promote global translation and upregulation of the Polycomb Respressor complex 2 (PRC2). Furthermore, the authors proved the crucial role of PRC2 in tumorigenesis on ErbB2-driven breast cancer model. Although the authors provided a strong in vivo data that highlights the relevance of c-Src/mTORC1/PRC2 axis in mammary tumors, the mechanism linking c-Src and mTORC1 could be strengthened by performing some additional experiments.

We thank the reviewer for their interesting and insightful comments. The requested experiments have indeed added important mechanistic details to the revised manuscript.

General comment:

In general, the manuscript is well presented and is easy to follow the logic behind the experiments. However, I suggest the authors include some additional references in the discussion to compare their results with data previously published.

In addition, I suggest correcting the format for some of the references.

We have added the requested references and amended the discussion. The format of all references has been checked and corrected where necessary.

Specific comments:

1. Line 113 please amend “tumourwe”

We apologize for the typos identified here and in comments 4-6 below and thank the reviewer for pointing them out. We have carefully checked the revised manuscript to ensure the absence of such errors.

2. Regarding Supplementary Figure 1d, There is some evidence showing that the c-Src play a role in angiogenesis (see articles to be included in the discussion). Considering this, the authors could show an endothelial cells marker in c-Src-proficient and c-Src-deficient tumors. This is important to understand to which extent the lack of c-Src is affecting the environment and how the latter is contributing to tumor growth in this model. This point could be also included in the discussion (line 355).

We agree with the reviewer that, even though the deletion of c-Src is limited to mammary epithelial/tumour cells in this model, cross-talk between tumour cells and the microenvironment may contribute to the phenotypes we observed. To assess angiogenesis specifically, we used quantitative immunohistochemistry to identify endothelial cells expressing the specific marker CD31. While this approach revealed no differences in angiogenesis between c-Src-deficient and control tumours (Supplementary Fig. 1F), further analysis of the tumour microenvironment in this model will be an interesting future direction.

3. Line 169 and Supplementary Figure 3b. The impairment on the mTORC1 pathway to sense amino acid in the c-Src-deficient cells may be related to a decrease in the amino acids uptake. As a consequence, lower intracellular levels of amino acids may explain the lack of activation of mTORC1 after the addition of amino acids. Does c-Src deficiency affect the expression of amino acid transporters?. The authors should measure the intracellular levels of amino acids in the c-Src-proficient and c-Src-deficient tumors/cells to elucidate whether the inhibition of mTORC1 is related or not to a decrease on the intracellular levels of amino acids.

Strikingly, GC/MS analysis revealed reduced steady state levels of many amino acids in c-Src-deficient tumour cells compared to wild-type controls (Supplementary Fig. 6C). Importantly, these reductions are not sufficient to activate the amino acid starvation response via GCN2-eIF2 α signaling, as the levels of phosphorylated eIF2 α (Ser51) do not differ in either tumour tissue or cell lines (Supplementary Fig. 4A-B). We also found no major effect of c-Src ablation on the expression of transcripts encoding amino acid transporters, with relatively few such genes up-regulated (4) or down-regulated (2) in c-Src-deficient tumours compared to controls.

An interesting possibility is the use of amino acids as substrates for anaplerosis, in which case lower steady state levels of amino acids in c-Src-deficient cells may stem from utilization of amino acids for energy generation and biosynthesis. Further studies testing this through metabolomics are ongoing, but we trust that these studies are beyond the scope of the present manuscript.

In our original manuscript, we demonstrated reduced mTORC1 reactivation in c-Src-deficient cells following amino acid starvation and re-stimulation. A recent publication in Nature Communications has implicated c-Src in amino acid sensing through suppression of the GATOR1 complex, which negatively regulates mTORC1 activation under conditions of amino acid insufficiency¹. We now show that stimulation of mTORC1 by amino acids, serum/growth factors or glucose following starvation is impaired in the absence of c-Src (Supplementary Fig. 4D,E and F, respectively). These data argue that impaired mTORC1 activation in c-Src-deficient cells is not linked uniquely to amino acid levels or sensing, but rather is consistent with inactivation of mTORC1 by energy stress, acting through AMPK-dependent mechanisms (Fig. 6, Supplementary Fig. 7A) that impair the response to multiple nutrients or growth factors. Nonetheless, we do not claim to rule out possible contributions of reduced intracellular amino acid levels or impaired amino acid sensing to mTORC1 inactivation in c-Src-deficient cells. However, we have demonstrated that mitochondrial dysfunction, which activates AMPK, is a major contributor to this phenotype, perhaps in combination with other factors such as altered amino acid levels or sensing thereof. We have incorporated these ideas into the revised results and discussion.

4 Line 193 please amend “that that”.

Please see the response to comment 1, above.

5. Line 202 please amend “tumouralso”.

See above.

6. Line 211 please amend “tumourFinally”

See above.

7. 222 and 241 The authors showed that the phosphorylation of TSC2 (pThr1462) was not affected, something that was expected given that they did not see differences in phospho-AKT (Se473). However, authors should check other phosphorylation site on TSC2 (pSer1387) and Raptor (P-Ser792), which are both mediated by the AMPK and impinge on mTORC1 signaling. In addition, amino acids starvation has been described as an activator of AMPK via CaMKKbeta (Ca²⁺/calmodulin-dependent protein kinase kinase-beta, Ghislat G et al. J Biol Chem. 2012;287:38625–38636.). Is this contributing to

the activation of AMPK in addition to the increased levels of AMP?. These experiments will contribute to clarify and provide a stronger mechanistic link between AMPK and mTORC1 for this model.

To address the first part of this comment, we immunoblotted c-Src-deficient and control cell extracts to detect phosphorylation of the AMPK target sites on TSC2 (Ser1387) and Raptor (Ser792). The results indicate enhanced phosphorylation of these sites in the absence of c-Src (Supplementary Figure 7A). As the reviewer suggests, this is consistent with the observed activation of AMPK in c-Src-deficient cells and agrees with our other findings showing that mTORC1 suppression in the absence of c-Src involves AMPK.

To address the second point, we treated c-Src-deficient and control cells with STO-609, a well-characterized and highly specific inhibitor of CAMKK β ². We used immunoblotting to monitor AMPK phosphorylation on Thr172 (catalyzed by kinases including CAMKK β) and the phosphorylation of the specific AMPK substrate ACC (Ser79). Notwithstanding that a basal level of AMPK Thr172 and ACC Ser79 phosphorylation in wild-type cells was dependent on CAMKK β , the response of c-Src-deficient cells to STO-609 was variable, with either a slight reduction in AMPK activity or no effect, indicating that a significant proportion of AMPK activity is maintained in the absence of CAMKK β activity (Supplementary Fig. 9A). In contrast, we used RNAi to show that Lkb1, which phosphorylates and activates AMPK under conditions of energy stress, contributes to the basal level of AMPK activation in control ErbB2+ cells, as previously reported³, and also drives the increased AMPK activation in c-Src-deficient cells (Supplementary Fig. 9B). Overall, these results indicate the primary role of energy stress in AMPK activation in this system, but are also consistent with multiple inputs into AMPK activation, acting either in parallel or cooperatively⁴.

s

8. Line 234-237 the authors claim that the deficiency in c-Src leads to a metabolic rewiring. Is this a metabolic rewiring induced by the lack of c-Src or the global decrease in the translation decrease also the expression of electron transport chain proteins, which could increase glycolysis?. Authors should measure the levels of the proteins of the electron transport chain to understand if this is connected or not to the decrease in respiration. Is the inhibition of mTORC1 sufficient to decrease the expression of these proteins in this context?. These experiments will provide new insights regarding the decrease in the respiration of c-Src deficient cells.

We immunoblotted extracts of control and c-Src-deficient ErbB2+ cells to detect mitochondrial proteins dependent on mTORC1 for their synthesis (TFAM, ATP5O)⁵ as well as surveying representative components of each ETC complex using a commercially available antibody mixture (OXPHOS antibody cocktail – Abcam). As the representative subunit of complex IV detected by this mixture can be difficult to resolve, we also included a separate antibody against one of the subunits of complex IV (CoxIV). The results indicated that c-Src ablation is associated with reduced expression of known mTORC1-dependent mitochondrial proteins and

some other components of ETC complexes, particularly complexes I and IV (Supplementary Fig. 7B-C). Interestingly, comparison of mitochondrial DNA content revealed no significant difference between control and c-Src-deficient cells (Supplementary Fig. 7D). Overall, these data indicate that c-Src deficiency may not affect mitochondrial content, but induces defects due to impaired synthesis of mitochondrial proteins. These findings are consistent with the known effects of mTORC1 on mitochondrial biogenesis and function^{5,6} and provide important additional insight into regulation of mitochondrial function by c-Src, as the reviewer suggests. We note, however, that silencing AMPK in c-Src-null cells fully restores mTORC1 activity while only partially restoring OXPHOS (Fig. 6H). Furthermore, acute c-Src ablation or kinase inhibition triggers a rapid reduction in OXPHOS (Fig. 6C, Supplementary Fig. 8C-D). Collectively, these observations suggest that mitochondrial dysfunction occurs as a direct consequence of c-Src inhibition, with subsequent suppression of mTORC1 activity by AMPK activation due to energy stress. Reduced mTORC1-dependent synthesis of key mitochondrial proteins then further impairs OXPHOS, thereby exacerbating energy stress. We have added text to the results and discussion describing these findings and their relationship to our overall model.

Reviewer #2

Polycomb Repressor Complex 2 (PRC2) has been shown to act either as an oncogene or as a tumour suppressor depending on the cellular context and tumour type. Components of the PRC2 complex (SUZ12, EZH2) have been shown to be highly expressed in certain cancer types, particularly in ERBB2-positive breast cancers (which also have global enrichment of H3K27me3). However, no mechanisms accounting for upregulation of PRC2 components (EZH2 and SUZ12) have been reported so far, and there is no evidence for an effect of PRC2 in driving mammary tumorigenesis raising the possibility that the high levels of H3K27me3, EZH2, and SUZ12 might be only correlative and not causal. This paper aims to address these outstanding questions using pathophysiologically relevant approaches. This is a real tour de force.

By demonstrating a complex and original interplay between c-Src oncogenic signaling, metabolic reprogramming, and epigenetic regulation, the authors propose a mechanism for how PRC2 components (EZH2 and SUZ12) are stabilised through regulation of the mRNA translation, and how PRC2 participates in ERBB2-driven tumorigenesis. Mechanistically, the authors showed that c-Src ablation impairs ERBB2-driven mammary tumorigenesis, and that c-Src promotes mTORC1 activation. Ultimately, this results in an increase in global mRNA translation and an increase in Ezh2 and Suz12 protein levels. Importantly c-Src ablation does not result in changes in the phosphorylation status of key signaling molecules such as Akt, Erk, Egfr, Erbb2, and Erbb3, suggesting a non-canonical mechanism of mTORC1 activation. c-Src ablation was shown to cause a reduction in oxidative phosphorylation and ATP reduction, leading to the activation of AMPK (a potent inhibitor of mTORC1). Finally, after deciphering mechanism through

which Ezh2 is upregulated in Erbb2-driven breast cancer, the authors show that ablation of Ezh2 strongly inhibits tumour growth.

The experiments are well controlled and the article is well-written, with a very detailed material and methods section. Many aspects of the research are thorough and well-discussed e.g. a good characterisation of mammary gland development of Ezh2 deletion in the mouse strain used. However, the overall narrative of the paper is sometimes difficult to follow, as it attempts to cover a lot of complex topics, and the connections between these topics are not immediately apparent. A small schematic summary of the results, incorporated into the final figure of the paper, would assist with this understanding. The authors should address the questions below for the manuscript to become suitable for publication.

We thank the reviewer for their positive and constructive comments and for the thorough review of our manuscript. We agree that c-Src has a complex role in controlling PRC2 via metabolism and mRNA translation. To assist the reader we have added a schematic in Figure 8D illustrating the major findings of our paper.

Major points

1. Figure 3 addresses the reduction in mTORC1 activity that occurs with Src deletion. The authors claim that c-Src enhances mTORC1 activity, which in turn increases the synthesis of proteins including Ezh2 and Suz12. To support their claim, they show immunoblotting of downstream targets of mTORC1 phospho-Rps6 and phospho-Eif4ebp1; however, there is no western blot directly showing the total/phospho levels of mTORC1 components. The authors should perform a western blot showing total and phosphorylation status of mTORC1 components (e.g. mTOR pSer2448) in control NIC/c-src+/+ versus NIC/c-src+/+ to address this question. mTORC1 component blots should be added to figure 3a as well as figures 4d and 5h (to address whether AMPK activation results in mTORC1 inhibition and not only in inhibition of downstream targets of mTORC1).

We used immunoblotting to examine the protein levels of mTORC1 components and the status of phosphorylation sites (where specific antibodies are readily available) in c-Src-deficient and control tumour tissue (Supplementary Figure 4C), PDX models treated with Src family kinase and mTOR inhibitors (Supplementary Figure 5E) and ErbB2+ breast cancer cells treated with mitochondrial inhibitors or AMPK agonists (Supplementary Figure 7F). Phosphorylation of Raptor on Ser792 has also been assessed in response to a comment from reviewer 1 (Supplementary Figure 7A). The data indicate that the expression of mTORC1 components is unaffected by the genetic and pharmacological strategies used in this study, while phosphorylation of mTOR and PRAS40 is reduced by SFK and mTOR inhibitor treatment of

PDX models (Supplementary Fig. 5E) and by inhibitors of ETC complexes I and V in MMTV-NIC tumour cell lines (Supplementary Fig. 7F – particularly for PRAS40).

Overall, our data are consistent with inhibition of mTORC1 via energy stress and AMPK, which occurs through mechanisms that are not known to involve changes in the expression of mTORC1 components but do involve phosphorylation of Raptor, which we show is elevated in c-Src-deficient cells (Supplementary Fig. 7A).

2. DNA microarrays and analysis (Figure 1d): the focus is purely on the up-regulated genes in the Src-deleted tumours. There is no discussion of the down-regulated genes, even though there are in fact more down-regulated than upregulated genes. There is a lack of functional analysis of the up/down-regulated genes (with the exception of glucose metabolism enrichment plot in figure 5g) – authors state that c-Src loss significantly impedes Erbb2-driven mammary tumorigenesis but there is no exploration of the pathways/functions affected by c-Src loss Are there more gene sets that are enriched that could explain the metabolic effects of c-Src deletion?

Our focus on up-regulated genes in c-Src-deficient tumours was because multiple modes of analysis (ChEA, ENCODE) converged on PRC2 as a key regulator of up-regulated genes (Fig. 1D). While the targets of PRC2 can show considerable cell-type specificity, this result led us to examine the possibility of loss of function of PRC2 in these tumours, which we extensively validated. In contrast, the same analyses of the down-regulated genes did not consistently identify any candidate regulatory factors (Supplementary Fig. 2C). In Supplementary Fig. 2A-B, we present functional analysis (KEGG, GO, REACTOME) of genes differentially expressed between c-Src-deficient and control ErbB2+ mammary tumours. Genes up-regulated in c-Src-deficient tumours are associated primarily with extracellular matrix and cell adhesion-related processes, while signatures related to central carbon metabolism (including glycolysis and mitochondrial metabolism) are also significantly represented. Functional analysis of genes down-regulated in c-Src-deficient tumours revealed fewer significantly affected pathways, despite the fact that more genes were down-regulated than up-regulated. While these pathways may be involved in the phenotypes of c-Src-deficient tumours, their role is not obvious and cannot be explored further within this manuscript. While we agree with the reviewer that adding further analysis of our gene expression data improves the paper, we would also like to clarify that we have extensively explored energy metabolism and mTORC1 activation as major functions dependent on c-Src in the models used here. The difficulty in fully ascribing the phenotypes of c-Src-deficient tumours to transcriptional changes may reflect the fact that c-Src affects transcription primarily through indirect mechanisms, or that the immediate effects of mTORC1 inactivation are on mRNA translation.

Similarly, for the H3K27me3 ChIP-seq (figure 1f) there is no functional analysis of the peaks that are regulated by c-Src expression: the peaks are nicely centred around the TSS and could be easily correlated with changes in gene expression. This would provide

another piece of evidence that the changes in gene expression are actually being caused by H3K27me3 loss due to a downregulation of PRC2 activity, and a functional readout of how c-Src alters cellular behaviour to promote tumorigenesis in Erbb2-driven tumours. At the moment the evidence provided for the gene expression changes being caused by PRC2 (downstream of c-Src deletion) is purely computational (ChEA and Encode analyses).

Analysis of protein-coding genes with H3K27me³ peaks within 20kb of their TSS in MMTV-NIC cells (Supplementary Fig. 2D) suggested that PRC2 target genes in this model are involved in functions related to neuronal differentiation, as we previously found in the PyV mT model, a breast cancer GEMM representing Luminal B breast cancer⁷. Interestingly, most of these genes were not expressed in c-Src-deficient cell lines (Supplementary Fig. 2F). This may suggest the engagement of alternative mechanisms to preserve their silencing. Indeed, we have preliminary evidence indicating that HDACs may be engaged in c-Src-deficient cells to maintain the transcriptional repression of genes normally targeted by PRC2, although this will require further validation including ChIP-Seq and functional studies which are beyond the scope of this manuscript. While the overlap between protein-coding genes silenced by PRC2 in control cells and genes transcriptionally up-regulated in c-Src-deficient cells is small (Supplementary Fig. 2F), a signature comprised of 40 such genes with H3K27me³ near their TSS exhibits a strong negative correlation with EZH2 protein expression in a cohort of human ERBB2+ breast cancer patients (Supplementary Fig. 2G), implying that these genes are bona fide PRC2 targets in this breast cancer subtype.

Perhaps most importantly, the vast majority of H3K27me³ peaks in cells from our ErbB2+ GEMM are found in non-coding regions of the genome (Supplementary Fig. 2E). This is in line with published studies showing that non-coding RNAs, genomic repeats and endogenous retroviruses are PRC2 targets in various cell types, and their re-expression upon loss of PRC2 function can have highly deleterious consequences⁸⁻¹⁰. Analyses focused on protein-coding genes are unable to capture the potentially crucial function of PRC2 in maintaining the silencing of certain non-coding RNAs, the de-repression of which may impair cell growth and survival and contribute to other transcriptional changes observed in c-Src-deficient tumours and cell lines. As quantification of non-coding transcripts involves specialized protocols, they cannot be analyzed using our current datasets. However, as we discuss in the revised manuscript, this is an interesting area for future study.

Although c-Src-deficient cells exhibit transcriptomic signatures of PRC2 identified in other cell types (Fig. 1G, original manuscript), we have no evidence that these genes are direct PRC2 targets in cell lines derived from the MMTV-NIC model. We have therefore removed Fig. 1G from the manuscript. Conceivably, we might have identified a larger correlation between mRNA expression and H3K27me³ by using tumour tissue samples, where we initially identified PRC2 signatures in c-Src-deficient tumours (Fig. 1D). However, because H3K27me³ is abundant in stromal cells in breast cancer GEMMs⁷ (also unpublished observations) and c-Src-null tumours

are necrotic (Supplementary Figure 1D), we used cell lines established from the GEMM, a defined system maintaining the proliferative and metabolic phenotypes observed *in vivo*, for epigenomic studies. Nonetheless, the absence of appropriate microenvironmental cues and the adaptation of the cells to culture conditions may alter the genomic landscape of PRC2 targets compared to tumour cells *in vivo*. Future studies, e.g. using FACS to isolate cells from pre-malignant lesions and tumours, followed by RNA-Seq and ChIP-Seq, will address these issues but are technically challenging and time-consuming and therefore cannot be included here. We emphasize that our ChIP-Seq analysis confirms that c-Src ablation drastically reduces PRC2 activity, which is strongly supported by evidence obtained using independent techniques (Fig. 1E, Fig. 2, Supplementary Fig. 3). While we agree that functional characterization of PRC2 genomic targets involved in mammary tumourigenesis is a worthy topic of study, our emphasis here is on the upstream regulation of PRC2 by tyrosine kinase signaling, energy metabolism and mRNA translation, which was previously undescribed. Further investigation of coding and non-coding PRC2 targets in breast carcinogenesis is therefore beyond the scope of our study and would be best served as the focus of a separate manuscript.

Are there genes that change in expression that are not correlated with H3K27me3 loss that could explain some of the driving metabolic effects of c-Src deletion?

Based on our gene expression and ChIP-Seq analysis, most genes with altered expression in c-Src-deficient cells are not direct targets of PRC2. We found that transcriptional signatures of metabolic pathways are almost exclusively up-regulated in c-Src-deficient cells and tumours. Prominent among these is glucose/pyruvate metabolism, as we discuss in our manuscript, although there are additional pathways related to mitochondrial function and lipid metabolism, among others (Supplementary Fig. 2A-B). This transcriptional regulation may occur as tumours evolve with impaired mitochondrial function in the absence of c-Src, making this an adaptive response that enables survival and a limited degree of proliferation. Our data suggest that the driving metabolic effects of c-Src deletion are more likely due to loss of c-Src-dependent tyrosine phosphorylation events and/or interactions that affect proteins regulating metabolism. In this case, while effects on transcription are likely to be important in determining the phenotype of c-Src-deficient tumours, they occur as an indirect consequence of c-Src loss. We clarified these points in the revised version of the manuscript.

Does re-expression of PRC2 sub-units such as Ezh2 and Suz12 rescue a subset of the gene expression changes?

To test this, we would need to express both of these subunits at or near endogenous levels, to preserve the stoichiometry of PRC2. While this may be possible, it would certainly be technically very challenging. For the reasons discussed above, it is also unlikely that the cell lines available to us would be suitable for these experiments, since they appear to have adapted to maintain the silencing of many genes that are targets of PRC2 in wild-type cells. Furthermore, since c-

Src controls PRC2 expression via post-transcriptional mechanisms involving mRNA translation, overexpressing PRC2 subunit mRNAs may not be entirely relevant as a rescue strategy.

3. In figure 1, the authors show that Src deletion impairs ErbB2-driven tumorigenesis; similarly, in the last figure they show that ablation of Ezh2 decreases ErbB2-driven tumorigenesis. In the title of the paper the authors claim that ErbB2/c-Src axis drives mammary tumorigenesis via metabolic reprogramming and translational up-regulation of PRC2, suggesting a synergistic contribution of c-Src and PRC2 to tumour development.

We did not intend for the title of our manuscript to imply that c-Src and PRC2 synergize to promote mammary tumourigenesis. Rather, our intention was to convey that metabolic reprogramming due to c-Src activity leads directly to translational upregulation of PRC2 components, which promotes tumourigenesis. Thanks to the reviewer, we now realize that the title of our manuscript was ambiguous and have elected to change it to “An ErbB2/c-Src axis drives mammary tumorigenesis through metabolically-directed translational regulation of Polycomb Repressor Complex 2“. We hope that the title conveys to the reader the previously unknown link between c-Src activity and PRC2 through the metabolic reprogramming-driven and mRNA translation-dependent mechanism that we have discovered.

Does treatment of Erbb2-driven tumours with dual-targeting therapy (e.g. dasatinib + Ezh2 inhibitor) have an added benefit over monotherapy (where resistance might be expected to evolve)? Can the authors perform this experiment using the ErbB2-driven breast cancer line derived from NIC model (same setting as figure 8b) ?

As stated above, our data do not suggest that c-Src and Ezh2 synergize, but rather that c-Src activity promotes Ezh2 overexpression. This may indicate that combination therapy against these two targets may not be the most effective strategy. However, tumours such as ErbB2+ breast cancer that express high levels of Ezh2 may require higher doses of Ezh2 inhibitors to achieve efficacy, raising the risk of toxicity, particularly with long-term treatment. To test the effect of reducing Ezh2 levels on sensitivity to Ezh2 inhibitors, we analyzed the *in vitro* proliferation of c-Src-deficient and control MMTV-NIC cells treated with the Ezh2 inhibitor GSK126. While the proliferation of wild-type ErbB2+ breast cancer cells was significantly reduced by Ezh2 inhibition, there was no significant difference in the proliferation of c-Src-deficient cells (Supplementary Figure 10D), suggesting that the low level of PRC2 activity remaining in c-Src-deficient cells is dispensable. While these findings do not preclude the possibility that co-targeting of c-Src and Ezh2 may be effective, we cannot currently justify the additional time and expense required to perform further *in vivo* studies as in Figure 8B. Furthermore, given the prominence of ErbB2-targeted therapy in this subtype of breast cancer, combinations of ErbB2 inhibitors with drugs targeting c-Src or Ezh2 are likely to be more clinically relevant, especially in the setting of resistance to ErbB2-targeted therapies where Src family kinases are highly active¹¹. These studies, which are in progress, would substantially

expand the scope of the present manuscript, beyond what is feasible, and will therefore form the basis of a following study.

4. Not fully addressed is how the metabolic effects of c-Src deletion, which appear to be the kick-start for the whole process of PRC2 inactivation, arise. e.g. is it due to changes in expression of metabolic genes (as demonstrated for glucose metabolism)? What transcription factors downstream of c-Src could be driving these gene expression changes? It is stated in the discussion that the precise mechanisms by which c-Src alters mitochondrial function have not been fully elucidated by other studies, could this point be further elaborated on?

We agree with the reviewer that alterations in transcriptional programs related to metabolic pathways in c-Src-deficient tumours (see response to comment 2) are likely to be important as adaptations that enable cell survival and/or as part of the overall metabolic impairment induced by c-Src loss. In Supplementary Fig. 2C we present an integrative analysis of potential transcriptional regulators of genes differentially expressed between c-Src-deficient and control tumour tissue and cell lines, based on multiple methods (ChEA, ENCODE, IPA) applied to both sample types. This confirms that genes with elevated expression in the c-Src-deficient models are regulated by PRC2 and additionally identifies signatures of known pro-tumourigenic factors (STAT3, CTNNB1), tumour suppressors (TP53), and factors implicated in stress responses (NFE2L2), and metabolism (CEBPB, PPARG, EBF1). The identification of signatures of SMARCA4 in these data is also interesting given the significant reduction in PRC2 activity in c-Src-null tumours and the evidence for mutual antagonism between PRC2 and SWI/SNF complexes in many cancers¹²⁻¹⁴. The same approach provided no information on potential regulators of genes with reduced expression in c-Src-deficient tumours compared to controls.

Overall, this analysis confirms that genes over-expressed in our ErbB2-driven model in the absence of c-Src are enriched in known PRC2 targets and also identifies other potential regulators of their expression that can be interrogated further in future studies. However, transcriptomic analyses did not identify any programs or pathways that clearly explain the loss of mitochondrial function induced by c-Src ablation. The relatively rapid effect of acute c-Src ablation or inhibition on mitochondrial function (Fig. 6, Supplementary Fig. 8) is more consistent with effects on mitochondria via signaling, rather than a more indirect effect mediated through transcription. In the discussion, we clarified these points as well as referring to published work identifying putative mitochondrial substrates of c-Src. This area of research is contentious, given that c-Src has no mitochondrial targeting sequence and mechanisms mediating regulation of mitochondria by c-Src and other kinases are incompletely defined. Interestingly, a recent study used metabolomic and proteomic approaches to identify a novel substrate of the non-receptor tyrosine kinase c-Abl on the mitochondrial outer membrane¹⁵, possibly representing a more plausible mechanism for kinase-dependent regulation of mitochondrial function. We are currently pursuing analogous strategies to identify novel c-Src targets and interactors regulating metabolism in breast cancer cells. As this will potentially require years of work for target

identification and validation, these studies are beyond the scope of this manuscript. However, in the discussion we have modified the text to improve the description of mitochondrial regulation by kinases such as c-Src.

Additional points

5. Add legend to figure 1d to make it clear which genes are up/down-regulated.

The legend for Figure 1D has been modified to address this request.

6. Relatively small number of H3K27me3 peaks even in the control setting (figure 1f) – how does this compare to other H3K27me3 ChIPs?

The genomic landscape of H3K27me³ is highly variable between cell types. It would therefore be most informative to compare our ChIP-Seq data with other data derived from mammary epithelial or breast cancer cells, although breast cancer is very heterogeneous and relatively few studies have specifically examined ErbB2+ breast cancer. However, examination of data from a range of breast cancer studies performing genome-wide profiling of H3K27me³ revealed that the number of peaks we identified in wild-type ErbB2+ tumour cells (average of 4245 peaks) is well within the typical range, e.g. 1000-5000 peaks for ErbB2+ human breast cancer cell lines¹⁶.

7. In relation to figure 2 – the authors state there is “globally reduced translation in Src-deficient cells”

a. However, there is no alteration in the polysome distribution for Eed and Actb; this suggests an element of selectivity for the effects on translation, rather than a global reduction, which is not addressed

The global reduction in mRNA translation is seen in the polysome profiles (Fig. 2D), where the c-Src-deficient cells (red traces) exhibit fewer heavy polysomes than controls (blue traces). This reduction in mRNA translation is consistent with mTORC1 inhibition and is in line with other studies where polysome profiling was applied to cells following inhibition of mTORC1^{17,18}. Although the reduced translation of mTORC1-dependent mRNAs is sufficient to shift the overall polysome profile, it still represents only a fraction of cellular mRNAs. Many mRNAs do not require mTORC1 for efficient translation, as is known for *Actb* and as we find here for *Eed*. Thus, the reviewer is correct in saying that these effects on mRNA translation are selective.

b. Why/how is the translation of some transcripts and not others affected?

c. E.g. do they bear a 5'TOP sequence (cytosine at the penultimate nucleotide position followed by a stretch of 4–14 pyrimidines) rendering them particularly sensitive to mTORC1 regulation ? This could also be addressed in the discussion

The variable effects of c-Src ablation on translation of particular mRNAs is consistent with mTORC1 inactivation, which affects the translation of a subset of mRNAs that includes those with 5'TOP sequences, but is not limited to mRNAs bearing this element. Indeed, recent studies have identified “non-TOP” mRNAs with both long and short 5'UTRs that are dependent on mTORC1 for efficient translation¹⁹. Thus, the presence or absence of a 5'TOP sequence is not the only factor determining whether translation of a given mRNA is mTORC1-dependent. Further complicating this issue is the frequent occurrence of alternative 5'UTRs for most genes, typically due to alternative transcription initiation sites, which can vary in a cell type- and context-specific manner. Techniques such as CAGE/nanoCAGE are required for comprehensive characterization of 5'UTRs in a given experimental system¹⁹. While this would be extremely interesting, it is beyond the scope of the study presented here.

8. Figure 3c – state in the figure or legend which ERbB2 cell line was used for this experiment

These were cell lines derived from the ErbB2+ GEMM used in our study (the MMTV-NIC model). We have clarified this in the legend for Figure 3.

9. The lack of phosphorylation changes for key signaling mediators (e.g. RTKs, Erk, Akt) with Src deletion would be better in a main figure, and if possible introduced earlier to make it clear that this canonical mechanism is not what is driving mTORC1 inactivation with Src deletion

We agree that the lack of effect of c-Src ablation on major RTK-dependent pathways in ErbB2+ tumours is interesting. We decided to place these observations in a supplementary figure mainly due to space constraints. As the revised manuscript is longer than the initial submission, this remains an issue and we have therefore decided it would be best to keep these data in a supplementary figure.

10. Figure 4d – can the authors add an immunoblot of H3K27me3 using the same extract to see whether c-Src and mTORC1 inhibition also result in a reduction in H3K27me3 in the PDX setting?

To analyze H3K27me³ levels in PDX models treated with inhibitors, we applied the immunohistochemical approach used in Fig. 1E. Consistent with the effects of drug treatments on EZH2 expression, inhibition of Src family kinases or mTOR significantly reduced H3K27me³ levels, with a larger proportion of cells expressing low levels (1+) or undetectable (0) H3K27me³

and a smaller proportion expressing high levels of H3K27me³ (3+) compared to controls (Supplementary Fig.5F).

11. Figure 5d – can the authors specify in the text that pT172 is a sign of AMPK activation?

We have added text to the results section specifying that phosphorylation of Thr 172 is required for AMPK activation above a very low baseline level.

12. Need to explain further page 11, line 247 - “the biguanide phenformin, which bypasses the input of c-Src”

We have clarified this passage of the text to explain that phenformin, by inhibiting Complex I, directly suppresses OXPHOS regardless of the status of c-Src. The data in Figure 5H therefore show that OXPHOS is required for PRC2 subunit overexpression in ErbB2+ breast cancer cells. This is consistent with our model whereby modulation of mitochondrial function mediates the effect of c-Src on PRC2 protein levels through signaling energy sufficiency to mTORC1.

13. In relation to figure 6:

a. Total AMPK and phosphor-AMPK levels should be shown in panel a

We have added immunoblots showing total and Thr172-phosphorylated AMPK α in this experiment (Fig. 6A).

b. In panel c, why do the c-Src L/L-LacZ cells have lower basal respiration than their +/- counterparts?

These cell lines are established from independent primary tumours that formed in different mice. While the mice are of the same transgenic strain (MMTV-NDL²⁰), we have found that cell lines derived from their tumours exhibit heterogeneity in many respects, including their basal metabolism. We observed that the basal metabolic properties of the control and floxed-Src cell lines we derived were different (although the floxed-Src cells appear to have a similar maximal respiratory capacity when treated with FCCP). However, the key point of this figure is the difference in the behavior of these cells upon introduction of Cre. Compared to their respective baselines, control cells exhibited no significant change, while cells with floxed Src alleles exhibited considerably reduced respiration.

14. In the text, line 317, the authors want to adress whether EZH2 ablation is required for tumor initiation however they do not provide data to answer this question

a. Can the author perform a tumorsphere assay (in Ultra low attachment plate or in 3D matrigel) to see whether EZH2 ablation impacts tumor initiating capacities of HER2 driven breast cancer cell derived from the NIC model?

b. Indeed, in the last figure, the delay in tumor growth is more likely due to a decreased proliferation (low Ki67 index) rather than inhibition in tumor initiation

We apologize for being insufficiently clear in the text describing these results. The reviewer is correct in saying that the phenotype of Ezh2-deficient mammary glands expressing ErbB2 is due to a proliferative block rather than a block in transformation *per se*. This is evident in the presence of hyperplastic lesions expressing ErbB2, but not Ki67, in Ezh2-deficient mammary glands. This block in tumour progression is quite profound, as escape is very infrequent, leading to an 80% reduction in mammary tumour penetrance in this normally very aggressive model. We have clarified the text to state that Ezh2 ablation does not block transformation but affects tumour progression, with arrest at an early hyperplastic stage. Experiments involving inhibition and genetic ablation of Ezh2 in organoid models are ongoing but will require significant further investments of time and resources.

Minor issues noted in abstract and introduction:

o Line 41 : « However the underlying mechanisms are obscure. » Mechanisms of what ? reformulate

o Line 45 : « c-src coordinates bioenergetics with PRC2 overexpression » what does this mean ? Description is vague and does not assist with understanding this complex pathway

o Line 68/69 : among the most prominent tumor type..... is breast cancer : reformulate

o Line 87 : upon directly by : reformulate

o Line 92/93/94 : not easily understood (provide an example of key signalling intermediates?)

We have modified the text to clarify each of these points.

o Line 55 : lack of reference highlighting a role of PRC2 in pluripotency

o Line 84 : lack of reference linking epigenetic dysregulation and metabolic reprogramming

o There is a heavy reliance on review articles rather than original research articles to support claims regarding the consequences of metabolic reprogramming on tumorigenesis e.g. lines 83-96

We have added references to original research articles describing the role of PRC2 in pluripotency and differentiation as well as the role of metabolic alterations in contributing to epigenetic dysregulation. In attempting to keep the number of cited publications at a reasonable number (the journal suggests 70 references), we have cited some reviews to limit the number of references. We agree that it is preferable to cite original research as much as possible, however for an area as broad as cancer metabolism too many citations of individual papers would be required to encapsulate the current position of the field. We agree that this is regrettable and we mean no disrespect to our colleagues whose work we cannot cite due to space limitations.

- **Spelling/grammatical errors:**
 - o **Line 113 – remove “tumour”**
 - o **Line 121 – remove open bracket**
 - o **Line 202 – remove “tumour”**
 - o **Line 211 – remove “tumour”**

We thank the reviewer for pointing out these errors, which we have now corrected.

In summary, we thank the reviewers for their efforts to improve our manuscript and trust that our responses have addressed their points adequately. I look forward to hearing from you again in the near future.

REFERENCES

- 1 Pal, R., Palmira, M., Chauhury, A., Klisch, T.J., di Ronza, A., Neilson, J. R., Rodney, G.G. and Sardiello, M. Src regulates amino acid-mediated mTORC1 activation by disrupting-Rag GTPase interaction. *Nature Comms.* **9**, 4351 (2018).
- 2 Tokumitsu, H. *et al.* STO-609, a specific inhibitor of the Ca(2+)/calmodulin-dependent protein kinase kinase. *The Journal of biological chemistry* **277**, 15813-15818, doi:10.1074/jbc.M201075200 (2002).
- 3 Dupuy, F. *et al.* LKB1 is a central regulator of tumor initiation and pro-growth metabolism in ErbB2-mediated breast cancer. *Cancer & metabolism* **1**, 18, doi:10.1186/2049-3002-1-18 (2013).

- 4 Fogarty, S. *et al.* Calmodulin-dependent protein kinase kinase-beta activates AMPK without forming a stable complex: synergistic effects of Ca²⁺ and AMP. *The Biochemical journal* **426**, 109-118, doi:10.1042/bj20091372 (2010).
- 5 Morita, M. *et al.* mTORC1 Controls Mitochondrial Activity and Biogenesis through 4E-BP-Dependent Translational Regulation. *Cell Metabolism* **18**, 698-711, doi:<https://doi.org/10.1016/j.cmet.2013.10.001> (2013).
- 6 Morita, M. *et al.* mTOR coordinates protein synthesis, mitochondrial activity and proliferation. *Cell cycle (Georgetown, Tex.)* **14**, 473-480, doi:10.4161/15384101.2014.991572 (2015).
- 7 Hirukawa, A. *et al.* Targeting EZH2 reactivates a breast cancer subtype-specific anti-metastatic transcriptional program. *Nat Commun* **9**, 2547, doi:10.1038/s41467-018-04864-8 (2018).
- 8 Walter, M., Teissandier, A., Pérez-Palacios, R. & Bourc'his, D. An epigenetic switch ensures transposon repression upon dynamic loss of DNA methylation in embryonic stem cells. *eLife* **5**, e11418, doi:10.7554/eLife.11418 (2016).
- 9 Ishak, C. A. *et al.* An RB-EZH2 Complex Mediates Silencing of Repetitive DNA Sequences. *Molecular cell* **64**, 1074-1087, doi:10.1016/j.molcel.2016.10.021 (2016).
- 10 Canadas, I. *et al.* Tumor innate immunity primed by specific interferon-stimulated endogenous retroviruses. *Nature medicine* **24**, 1143-1150, doi:10.1038/s41591-018-0116-5 (2018).
- 11 Rexer, B. N. *et al.* Phosphoproteomic mass spectrometry profiling links Src family kinases to escape from HER2 tyrosine kinase inhibition. *Oncogene* **30**, 4163-4174 (2011).
- 12 Erkek, S. *et al.* Comprehensive Analysis of Chromatin States in Atypical Teratoid/Rhabdoid Tumor Identifies Diverging Roles for SWI/SNF and Polycomb in Gene Regulation. *Cancer cell*, doi:10.1016/j.ccell.2018.11.014 (2018).
- 13 Kim, K. H. *et al.* SWI/SNF-mutant cancers depend on catalytic and non-catalytic activity of EZH2. *Nature medicine* **21**, 1491-1496, doi:10.1038/nm.3968 (2015).
- 14 Januario, T. *et al.* PRC2-mediated repression of SMARCA2 predicts EZH2 inhibitor activity in SWI/SNF mutant tumors. *Proceedings of the National Academy of Sciences of the United States of America* **114**, 12249-12254, doi:10.1073/pnas.1703966114 (2017).
- 15 Kurmi, K. *et al.* Tyrosine Phosphorylation of Mitochondrial Creatine Kinase 1 Enhances a Druggable Tumor Energy Shuttle Pathway. *Cell Metab* **28**, 833-847.e838, doi:10.1016/j.cmet.2018.08.008 (2018).
- 16 Xi, Y. *et al.* Histone modification profiling in breast cancer cell lines highlights commonalities and differences among subtypes. **19**, 150, doi:10.1186/s12864-018-4533-0 (2018).

- 17 Hsieh, A. C. *et al.* The translational landscape of mTOR signalling steers cancer initiation and metastasis. *Nature* **485**, 55-61, doi:10.1038/nature10912 (2012).
- 18 Larsson, O. *et al.* Distinct perturbation of the translome by the antidiabetic drug metformin. *Proceedings of the National Academy of Sciences of the United States of America* **109**, 8977-8982, doi:10.1073/pnas.1201689109 (2012).
- 19 Gandin, V. *et al.* nanoCAGE reveals 5' UTR features that define specific modes of translation of functionally related MTOR-sensitive mRNAs. *Genome research* **26**, 636-648, doi:10.1101/gr.197566.115 (2016).
- 20 Siegel, P. M., Ryan, E. D., Cardiff, R. D. & Muller, W. J. Elevated expression of activated forms of Neu/ErbB-2 and ErbB-3 are involved in the induction of mammary tumors in transgenic mice: implications for human breast cancer. *The EMBO journal* **18**, 2149-2164, doi:10.1093/emboj/18.8.2149 (1999).

REVIEWERS' COMMENTS:

Reviewer #1 (Remarks to the Author):

The manuscript by Smith HW et al, now entitled "An ErbB2/c-Src axis drives mammary tumorigenesis through metabolically directed translational regulation of Polycomb repressor complex2", have been improved by the addition of new experimental data that strengthened the conclusions of the study. In the revised manuscript, the authors provided a more detailed molecular mechanism to explain the inhibition of mTORC1 in the cells lacking c-Src. The activation of AMPK elicited by the absence of c-Src, lead to the inhibition of mTORC1, which in turn led to a decrease in respiration, lower expression of PRC2 and ultimately to the impairment of tumor growth.

Altogether, the in vivo and in vitro data provided now by the authors converge in a more complete molecular mechanism that highlights the role of c-Src/mTORC1/PRC2 in breast cancer. Finally, the novelty and quality of the manuscript make it suitable for publication in Nature Communications.

Specific comments:

Page 10. The first paragraph should say supplementary fig.4c instead of Supplementary fig. 4b, please amend.

Supplementary figure 10 title says Supplementary figure 8, please amend.

Reviewer #2 (Remarks to the Author):

The authors performed an important number of experiments that increase the quality of the manuscript. It is now acceptable for publication.

SRC ablation decreased phosphorylation of downstream targets of mTORC1 (P-Rps6 and P-Eif4bp1), and the authors conclude that this is due to a decreased mTORC1 activity in Src -/- settings. Then they investigate the phosphorylation of different mTORC1 components, including mTOR itself, Raptor and PRAS40 in control versus Src -/- settings and found no changes (sup figure 4C). How do they explain that the phosphorylation status of the different mTORC1 components remains unchanged ? (raptor phosphorylation is even increased in SRC -/-, sup figure 7A). The authors should comment on/discuss this point.

Since the authors found that src inhibition using dasatinib inhibits mTOR phosphorylation, they should discuss why Src ablation does not recapitulate this result?

Reviewer #3 (Remarks to the Author):

In the revision of manuscript, the authors addressed the majority of reviewers' concerns and make the article more solid and comprehensive. My only suggestion is that the authors should avoid the use of terms like "globally reduced translation in Src-deficient cells" in Figure 2 related paragraphs. The authors have admitted in the response letter that it is more like a selective translation inhibition but not global translation inhibition. It will make the readers misunderstand that global translation inhibition will mainly affect the role of PRC2, but not other biological functions. Overall, I am impressed by the data size the authors have provided in this manuscript and it will definitely meet the standard to be published in NC after minor revision.

Reviewer #1

Specific comments:

Page 10. The first paragraph should say supplementary fig.4c instead of Supplementary fig. 4b, please amend.

Supplementary figure 10 title says Supplementary figure 8, please amend.

We thank the reviewer for identifying these discrepancies, which we have now amended.

Reviewer #2

SRC ablation decreased phosphorylation of downstream targets of mTORC1 (P-Rps6 and P-Eif4bp1), and the authors conclude that this is due to a decreased mTORC1 activity in Src -/- settings. Then they investigate the phosphorylation of different mTORC1 components, including mTOR itself, Raptor and PRAS40 in control versus Src -/- settings and found no changes (sup figure 4C). How do they explain that the phosphorylation status of the different mTORC1 components remains unchanged ? (raptor phosphorylation is even increased in SRC -/-, sup figure 7A). The authors should comment on/discuss this point.

We investigated phosphorylation of mTORC1 components in response to previous comments from the reviewers. We found that c-Src ablation elevated the AMPK-dependent phosphorylation of mTORC1 components (i.e. Raptor) and upstream regulators (TSC2). This is consistent with the phenotypes induced by c-Src knockout, including energy stress due to diminished mitochondrial function, which activates AMPK. These AMPK-dependent phosphorylation events are known to reduce mTORC1 activity. We also demonstrated that silencing AMPK catalytic subunits was sufficient to restore mTORC1 activity in c-Src-null cells. Therefore, our data argue that AMPK suppresses mTORC1 activity in c-Src-deficient cells, at least in part, through established mechanisms involving phosphorylation of Raptor and TSC2. The other phosphorylation sites on mTORC1 components that we investigated (mTOR Ser 2448 and PRAS40 Thr246) are targets of Akt. These sites were unchanged in c-Src-deficient

cells, which is consistent with the lack of effect of c-Src knockout on upstream signaling through ErbB2, which activates PI-3 kinase, leading to Akt activation. We show that phosphorylation of Akt on Ser473 is unaffected by c-Src knockout, as is the level of phosphorylation of other known Akt targets (TSC2 Thr1462) (Supplementary Figure 6A). The observed effects on phosphorylation of mTORC1 components are therefore in accordance with the other data presented in the paper. While the Akt phosphorylation sites on mTORC1 components are undoubtedly important, their alteration is not required for changes in mTORC1 activity to occur. Many factors can influence mTORC1 activation and our data, along with those of other studies, suggest that changes in metabolic inputs can suppress mTORC1 activity even in the context of active growth factor signaling. We have added text to the results and discussion to clarify our findings on phosphorylation of mTORC1 components in c-Src-deficient, ErbB2+ breast cancer cells.

Since the authors found that src inhibition using dasatinib inhibits mTOR phosphorylation, they should discuss why Src ablation does not recapitulate this result?

We agree with the reviewer that the differences between SFK inhibitor treatment and genetic c-Src ablation are interesting, although they are not entirely surprising. Dasatinib, in particular, is well-known to target a broad spectrum of kinases, including not only all 8 SFKs but also both isoforms of Abl and numerous receptor tyrosine kinases. We have used it in our experiments primarily because of its high potency and because it is the most frequently used SFK inhibitor in clinical trials. However, its lack of specificity can lead to effects on signaling that are not attributable directly to inhibition of c-Src. In the PDX models used in Figure 4 and Supplementary Figure 5, it is likely that Dasatinib targets other than c-Src contribute to the activation of PI-3K/Akt signaling, leading to a reduction in Akt-dependent phosphorylation of mTOR Ser2448 and PRAS40 Thr246 upon Dasatinib treatment. We have added text to the description of Supplementary Figure 5 in the Results section and to the Discussion to address these issues.

Reviewer #3

In the revision of manuscript, the authors addressed the majority of reviewers' concerns and make the article more solid and comprehensive. My only suggestion is that the authors should avoid the use of terms like "globally reduced translation in Src-deficient cells" in Figure 2 related paragraphs. The authors have admitted in the response letter that it is more like a selective translation inhibition but not global translation inhibition. It will make the readers misunderstand that global translation inhibition will mainly affect the role of PRC2, but not other biological functions. Overall, I am impressed by the data size the authors have provided in this manuscript and it will definitely meet the standard to be published in NC after minor revision.

We agree with the reviewer that terms such as “globally reduced translation” could potentially mislead readers. For clarity, in the sections of text referring to the results in Figure 2 we now refrain from describing the effects of c-Src knockout on translation in these terms and have instead emphasized that the shift in polysome profiles we observed (Figure 2d) is due to the selective loss of translation of a subset of mTORC1-dependent mRNAs, in agreement with other studies examining the effects of mTORC1 inhibition on translation.

In summary, through the feedback and suggestions of the reviewers we are now able to present a clearer, more comprehensive manuscript. We greatly appreciate your efforts and those of the reviewers towards improving our manuscript and preparing it for publication. We trust that the changes described here and in the cover letter have adequately addressed the remaining issues with this manuscript and look forward to proceeding with publication.